# Sororin actively maintains sister chromatid cohesion

Rene Ladurner[1,†], Emanuel Kreidl[1,†,‡], Miroslav P Ivanov[1], Heinz Ekker[2], Maria Helena Idarraga-Amado[1], Georg A Busslinger[1], Gordana Wutz[1], David A Cisneros[1] & Jan-Michael Peters[1,*]

## Abstract

Cohesion between sister chromatids is established during DNA replication but needs to be maintained to enable proper chromosome–spindle attachments in mitosis or meiosis. Cohesion is mediated by cohesin, but also depends on cohesin acetylation and sororin. Sororin contributes to cohesion by stabilizing cohesin on DNA. Sororin achieves this by inhibiting WAPL, which otherwise releases cohesin from DNA and destroys cohesion. Here we describe mouse models which enable the controlled depletion of sororin by gene deletion or auxin-induced degradation. We show that sororin is essential for embryonic development, cohesion maintenance, and proper chromosome segregation. We further show that the acetyltransferases ESCO1 and ESCO2 are essential for stabilizing cohesin on chromatin, that their only function in this process is to acetylate cohesin's SMC3 subunit, and that DNA replication is also required for stable cohesin–chromatin interactions. Unexpectedly, we find that sororin interacts dynamically with the cohesin complexes it stabilizes. This implies that sororin recruitment to cohesin does not depend on the DNA replication machinery or process itself, but on a property that cohesin acquires during cohesion establishment.

**Keywords** cell cycle; cohesin acetylation; mitosis; sister chromatid cohesion
**Subject Categories** Cell Cycle; DNA Replication, Repair & Recombination
**The EMBO Journal (2016) 35: 635–653**

## Introduction

The accurate segregation of replicated chromosomes during mitosis depends on the prior attachment of sister kinetochores to microtubules from opposing spindle poles. This bi-orientation is achieved by stabilization of microtubule–kinetochore attachments, which are exposed to tension generated by spindle forces (Maresca & Salmon,

2009; Uchida *et al*, 2009). Tension at kinetochores can only be generated because sister DNA molecules become physically connected during DNA replication. This sister chromatid cohesion resists spindle pulling forces, thereby enables the generation of tension, which stabilizes microtubule–kinetochore attachments (Dewar *et al*, 2004) and is therefore essential for chromosome bi-orientation. Once all chromosomes have become bi-oriented, cohesion is destroyed and sister chromatids are segregated by the mitotic spindle.

Sister chromatid cohesion is mediated by cohesin complexes (Guacci *et al*, 1997; Michaelis *et al*, 1997; Losada *et al*, 1998). The cohesin subunits SMC1, SMC3, and SCC1 (also called MCD1 and RAD21) form ring-like structures (Anderson *et al*, 2002; Haering *et al*, 2002; Huis in 't Veld *et al*, 2014), which are thought to mediate cohesion by topologically entrapping both sister DNAs (Haering *et al*, 2008). In metazoan cells, cohesin associates with chromatin already before DNA replication (Losada *et al*, 1998; Sumara *et al*, 2000) in a manner that depends on integrity of the cohesin ring (Pauli *et al*, 2008; Huis in 't Veld *et al*, 2014), implying that cohesin also interacts with unreplicated DNA via topological entrapment. These cohesin–DNA interactions can be reversed by the cohesin-associated protein WAPL (Gandhi *et al*, 2006; Kueng *et al*, 2006; Tedeschi *et al*, 2013), which releases cohesin from DNA by opening a DNA "gate" between SMC3 and SCC1 (Chan *et al*, 2012; Buheitel & Stemmann, 2013; Eichinger *et al*, 2013; Huis in 't Veld *et al*, 2014). Because cohesin can be continuously loaded onto DNA and released again by WAPL, cohesin–DNA interactions are dynamic before DNA replication (Gerlich *et al*, 2006). Similar cohesin–DNA interactions might exist in post-mitotic cells (Wendt *et al*, 2008) in which cohesin has roles in chromatin structure and gene regulation (reviewed in Seitan & Merkenschlager, 2012).

Cohesin establishes cohesion during DNA replication in yeast (Uhlmann & Nasmyth, 1998) and presumably also in mammals (Schmitz *et al*, 2007; Tachibana-Konwalski *et al*, 2010) and other eukaryotes. This process coincides with the acetylation of two lysine residues on SMC3 (K105 and K106 in human Smc3; Ben-Shahar *et al*, 2008; Unal *et al*, 2008; Zhang *et al*, 2008a) by the acetyltransferases ESCO1 and ESCO2 (orthologs of Eco1/Ctf7 in budding yeast;

1 IMP Research Institute of Molecular Pathology, Vienna, Austria
2 Campus Science Support Facilities, NGS Facility, Vienna, Austria
*Corresponding author. Tel: +43 1797303002; E-mail: Jan-Michael.Peters@imp.ac.at
†These authors contributed equally to this work
‡Present address: Massachusetts Institute of Technology, Cambridge, MA, USA
The copyright line of this article was changed on 26 August 2016 after original online publication

Hou & Zou, 2005; Ivanov *et al*, 2002; Skibbens *et al*, 1999; Toth *et al*, 1999). These modifications are required for cohesion (Ben-Shahar *et al*, 2008; Unal *et al*, 2008; Zhang *et al*, 2008a; Rowland *et al*, 2009; Sutani *et al*, 2009; Song *et al*, 2012; Whelan *et al*, 2012). In vertebrates, SMC3 acetylation results in the association of cohesin with sororin (Lafont *et al*, 2010; Nishiyama *et al*, 2010; Song *et al*, 2012). Like SMC3 acetylation, sororin is essential for cohesion (Rankin *et al*, 2005), but only in the presence of WAPL (Nishiyama *et al*, 2010), indicating that sororin's role in cohesion is to inhibit WAPL. Sororin is also required for stabilization of cohesin on chromatin (Schmitz *et al*, 2007), a process that occurs during DNA replication (Gerlich *et al*, 2006).

Based on these observations we have proposed that SMC3 acetylation is required for sister chromatid cohesion because it promotes the recruitment of sororin, which stabilizes cohesin on chromatin by inhibiting WAPL (Nishiyama *et al*, 2010). According to this hypothesis, SMC3 acetylation and sororin recruitment would not be essential for establishment of cohesion *per se*, but would instead be required to maintain cohesion, which could otherwise be destroyed precociously by WAPL before chromosomes have become bioriented on the mitotic spindle.

This hypothesis makes several important predictions, which we have tested here. Our results indicate that in human cells the SMC3 acetyltransferases ESCO1 and ESCO2 are required for stabilization of cohesin on chromatin during DNA replication, as one would predict if SMC3 acetylation leads to interactions between cohesin and sororin and to inhibition of WAPL. We provide evidence that the only function of ESCO1 and ESCO2 in cohesin stabilization is to acetylate SMC3. Interestingly, our results imply that sororin cannot stabilize acetylated cohesin on chromatin before DNA replication, consistent with the previous observation that SMC3 acetylation promotes sororin binding only when it occurs during DNA replication (Lafont *et al*, 2010; Nishiyama *et al*, 2010; Song *et al*, 2012). These observations indicate that binding of sororin to cohesin depends on both SMC3 acetylation and DNA replication. We provide further support for this hypothesis by showing that sororin is first recruited to cohesin on replicated DNA. Although sororin enables cohesin to reside on chromatin stably for many hours, sororin itself interacts with cohesin in a highly dynamic manner, as do sororin's antagonist WAPL and its binding partner PDS5A. This implies that it is not the presence of DNA replication proteins or the replication process *per se*, but some unknown property of cohesin on replicated DNA that enables sororin to interact with cohesin. Finally, by experimentally inducing the degradation of sororin either during or after DNA replication, we show that sororin is required for the maintenance of cohesion. These results support the hypothesis that sororin is recruited to acetylated cohesin complexes on replicated DNA in order to prevent the precocious release of these complexes from DNA by WAPL, a situation that is essential for the maintenance of cohesion until chromosomes have become bi-oriented on the mitotic spindle.

# Results

### ESCO1 and ESCO2 are required for stable association of cohesin with chromatin

If SMC3 acetylation is required for cohesion because it stabilizes cohesin on chromatin by enabling sororin recruitment and WAPL

inhibition (Nishiyama *et al*, 2010), the enzymes that acetylate SMC3 should also be required for stable cohesin–chromatin interactions in G2-phase. To test this, we depleted ESCO1 and ESCO2 by RNA interference (RNAi) from HeLa cells expressing SMC3 fused to green-fluorescent protein (GFP), synchronized these cells in G2-phase by release from a thymidine-induced DNA replication arrest (Fig 1A, Appendix Fig S1A), and measured cohesin–chromatin interactions in inverse fluorescence recovery after photobleaching (iFRAP) experiments (Fig 1B). The fluorescent version of SMC3 used in these experiments is a mouse protein, contains GFP as part of a C-terminal localization-affinity purification (LAP) tag (Poser *et al*, 2008), and is expressed from a stably integrated bacterial artificial chromosome (BAC). Previous experiments had shown that mouse SMC3-LAP expressed under these conditions is present at levels below endogenous SMC3 and assembles into functional cohesin complexes (Ladurner *et al*, 2014). Immunoblotting experiments revealed that most, although not all, ESCO1 and ESCO2 could be depleted by RNAi (Fig 1A). Immunoblotting with an antibody that specifically recognizes the acetylated form of SMC3 (Smc3(ac); Nishiyama *et al*, 2010) confirmed that both enzymes contribute to SMC3 acetylation (Fig 1A). Immunoblot analyses of proteins in chromatin fractions showed that depletion of ESCO1 and ESCO2 did not detectably change the steady-state levels of cohesin on chromatin, reminiscent of the situation in sororin-depleted cells (Schmitz *et al*, 2007; Nishiyama *et al*, 2010; van der Lelij *et al*, 2014).

By measuring the difference in SMC3 fluorescence intensity between unbleached and bleached areas of cell nuclei over time in iFRAP experiments, we observed an increase in cohesin's mobility after depletion of ESCO1 and ESCO2 (Fig 1C). Using a bi-exponential function to fit the decay curves allowed quantitative data interpretation (Appendix Fig S1). Similar to previous observations (Gerlich *et al*, 2006), we detected two populations of cohesin bound to chromatin (Fig 1D). In control cells, approximately 60% of cohesin complexes interacted with chromatin dynamically with a residence time in the range of 15–20 min, whereas the remaining 40% persisted on chromatin for at least 9–12 h. Two cohesin populations could also be detected in ESCO1- and ESCO2-depleted cells, but the ratio between these was altered. Cells depleted only of ESCO1 or ESCO2 showed small reductions in stably bound cohesin complexes to 33 and 22%, respectively, whereas in doubly depleted cells only 10% of cohesin was stably bound to chromatin. The calculated residence time of these residual stably bound cohesin complexes was not significantly altered by depletion of ESCO1 and ESCO2. A reduction in the number of stably bound cohesin complexes but not in their residence time on chromatin has previously also been observed in sororin-depleted cells (Schmitz *et al*, 2007; van der Lelij *et al*, 2014). These results show that ESCO1 and ESCO2 are required for stable binding of cohesin to chromatin in G2-phase and are consistent with the possibility that ESCO1 and ESCO2 mediate this effect by enabling cohesin–sororin interactions.

### SMC3 acetylation site mutants generate stable cohesin–chromatin interactions without ESCO1 and ESCO2

We tested next if SMC3 acetylation is sufficient for the stabilization of cohesin on chromatin and if the only function of ESCO1 and ESCO2 in this process is to acetylate SMC3. For this purpose, we

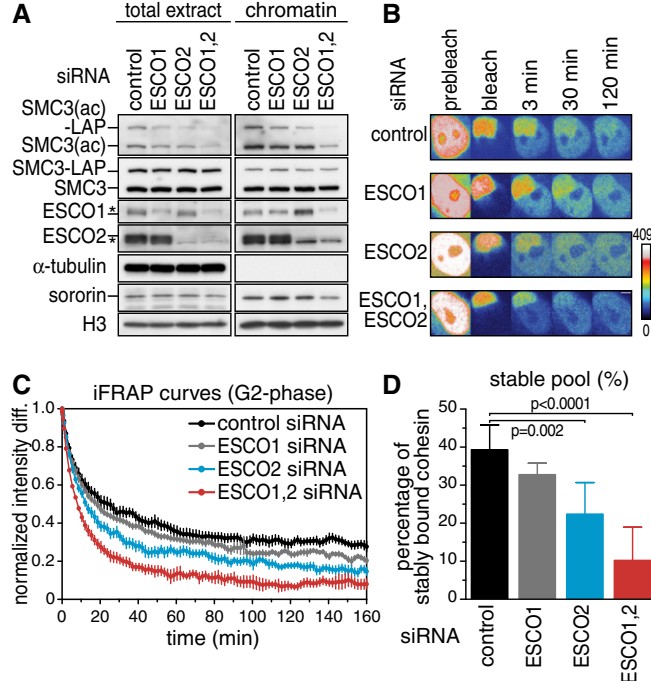

**Figure 1. ESCO1 and ESCO2 are required for stable association of cohesin with chromatin.**

A  Western blot of SMC3-LAP-expressing HeLa cells treated with siRNA as indicated and synchronized in G2-phase. Efficiency of RNA interference and SMC3 acetylation levels were analyzed from total extracts and after chromatin fractionation (asterisks indicate unspecific signals that were depleted from antibody dilutions in later experiments, presumably by repeated usage of the antibody samples). Note that ESCO1 levels were increased on chromatin after depletion of ESCO2, raising the possibility that a decrease in chromatin-bound ESCO2 might be compensated for by ESCO1 recruitment to chromatin.

B  Still images of an inverse fluorescence recovery after photobleaching (iFRAP) experiment. Fluorescence intensities are false colored as indicated. Scale bar, 5 μm.

C  Graph depicting the normalized intensity after photobleaching from two biological replicates to quantify chromatin dissociation kinetics. Error bars denote standard error of the mean (s.e.m.), n = 10 cells per condition.

D  Quantification of the relative abundance of SMC3-LAP in the stable chromatin binding mode. Error bars denote s.e.m.; unpaired *t*-test was used to compare conditions.

analyzed in iFRAP experiments the behavior of cohesin complexes in which K105 and K106 of LAP-tagged SMC3 had been mutated to either glutamine (Q) or arginine (R) residues. We had found previously that soluble cohesin complexes containing either this SMC3-LAP(QQ) or the SMC3-LAP(RR) mutant associate with sororin (Nishiyama *et al*, 2010). This is in contrast to wild-type cohesin, which can only associate with sororin if cohesin is associated with chromatin and if its SMC3 subunit has been acetylated (Lafont *et al*, 2010; Nishiyama *et al*, 2010). With respect to sororin binding, both mutants therefore resemble acetylated SMC3, despite the fact that arginine residues are structurally not particularly similar to acetylated lysine residues. As discussed by Rowland *et al* (2009), these mutations may therefore functionally resemble acetylated cohesins rather than mimic them structurally. We therefore refer to these as acetylation bypass mutants.

We first performed iFRAP experiments using cells synchronized in G1-phase, in which wild-type cohesin interacts with chromatin dynamically. The iFRAP recovery curves of both SMC3 mutants were similar to the one of wild-type SMC3-LAP (Fig 2A). All three curves could be fitted with a single exponential function, corresponding to a single pool of chromatin-associated cohesin with a residence time of 20 min (Fig 2B). Similar behavior of wild-type and mutant cohesin was also observed in cells synchronized in G2-phase, in which 40% of all wild-type cohesin complexes interacted with chromatin stably (Fig 1C). Also in these cells, the iFRAP recovery curves of both SMC3 mutants were similar to the one of wild-type SMC3-LAP (Fig 2C) and in this case indicated that 35–40% of both wild-type and mutated cohesin complexes were stably associated with chromatin (Fig 2D, Appendix Fig S2A and B). In other words, cohesin complexes containing mutations in SMC3 at the acetyl-lysine sites behaved exactly like wild-type cohesin in these assays. The observation that these mutant cohesin complexes do not stably associate with chromatin in G1-phase indicates that SMC3 acetylation is not sufficient for the stabilization of cohesin on chromatin, as was expected because sororin, which is degraded in G1-phase by the anaphase promoting complex (APC/C; Nishiyama *et al*, 2010; Rankin *et al*, 2005), is also required for this process (Schmitz *et al*, 2007).

We next wanted to address if the mutant alleles of SMC3 used here could support viability in the absence of endogenous SMC3. The coding sequences of the human and mouse homologs are 91.1% identical, and we failed to find conditions in which we could specifically deplete the human protein. We therefore introduced the aforementioned BAC constructs into mouse fibroblasts from a conditional *Smc3* "knockout" mouse model. Upon Cre-mediated deletion of endogenous *Smc3*, cells without SMC3-LAP stopped to proliferate, whereas cells containing wild-type SMC3-LAP, SMC3-LAP(QQ), or SMC3-LAP(RR) continued proliferation at a comparable rate. After two passages, the faster-migrating band corresponding to endogenous SMC3 could not be detected in immunoblotting experiments, whereas SMC3-LAP was expressed (Fig 2E), indicating that wild-type and mutant forms of SMC3-LAP supported cell viability.

We tested next if the presence of acetyl-mimicking residues in SMC3 enables cohesin to associate with chromatin stably in G2-phase in the absence of ESCO1 and ESCO2. Using HeLa cells, we depleted both enzymes by RNAi (Fig 2F, Appendix Fig S2C and D), synchronized cells in G2-phase, and analyzed the behavior of cohesin in iFRAP experiments (Fig 2G; Appendix Fig S2E and F). As observed before (Fig 1D), depletion of ESCO1 and ESCO2 reduced the abundance of stably chromatin-bound wild-type cohesin complexes from 35% to 10%. In contrast, depletion of ESCO1 and ESCO2 did not reduce the abundance of stably chromatin-bound SMC3-LAP(QQ) cohesin complexes. In a similar experiment, also stably chromatin-bound SMC3-LAP(RR) containing cohesin complexes were refractory to depletion of ESCO1 and ESCO2 (Appendix Fig S2G–I). This indicates that these mutants bypass the requirement for ESCO1 and ESCO2 and that the only function of ESCO1 and ESCO2 in stabilizing cohesin on chromatin is to acetylate SMC3.

However, as observed for wild-type cohesin (Schmitz *et al*, 2007; van der Lelij *et al*, 2014), sororin depletion reduced the number of cohesin complexes containing SMC3-LAP(QQ) which were stably associated with chromatin in G2-phase (Fig 2G). This finding and the observation that acetyl-bypass mutant cohesin complexes do not

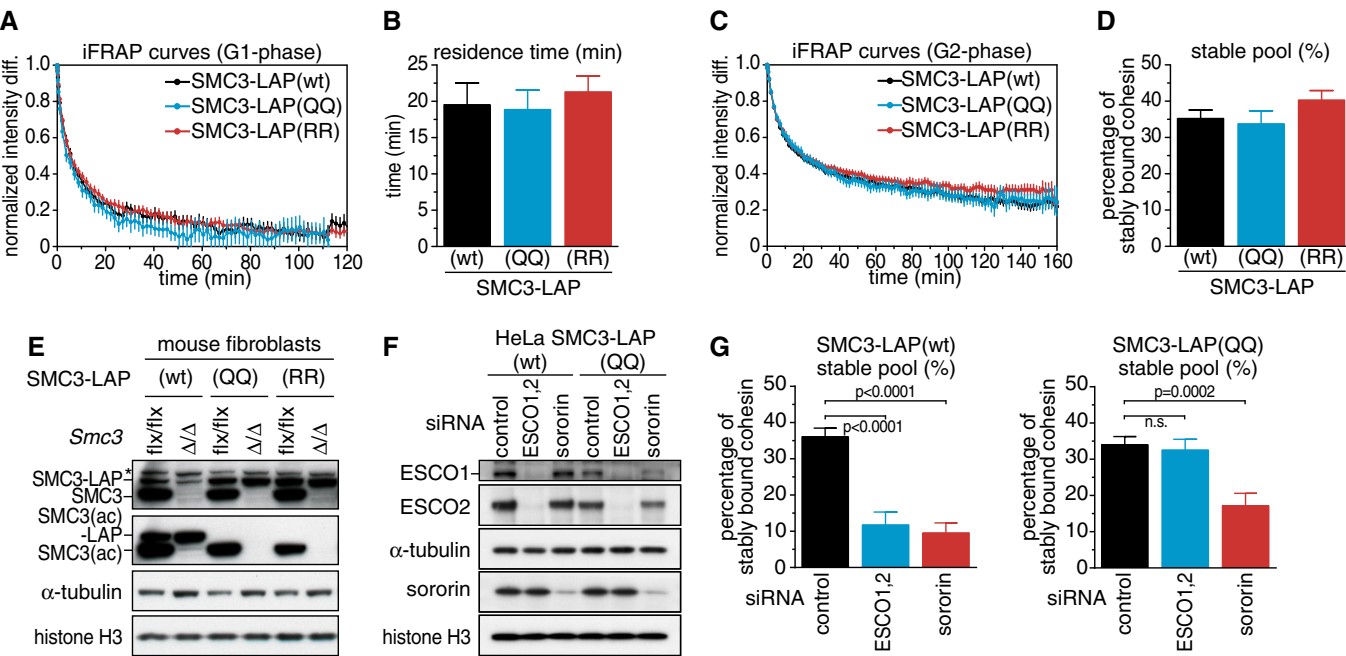

**Figure 2.** **SMC3 acetylation site mutants generate stable cohesin–chromatin interactions without ESCO1 and ESCO2.**

A   Quantification of iFRAP experiments using HeLa cells expressing SMC3-LAP in the wild-type form (wt) or with lysines 105-106 mutated to glutamines (QQ) or arginines (RR) and synchronized in G1-phase. Error bars denote s.e.m., *n* > 25 cells per condition.

B   Quantification of chromatin residence time of SMC3-LAP alleles in G1-phase. Error bars denote s.e.m.

C   Quantification of iFRAP experiments using cells synchronized in G2-phase. Error bars denote s.e.m., *n* > 26 cells per condition.

D   Quantification of stably bound SMC3-LAP alleles in G2-phase. Error bars denote s.e.m.

E   Western blot of mouse fibroblasts expressing SMC3-LAP in the presence or absence of endogenous *Smc3* after gene deletion. Extracts were prepared after subculturing two times. Cells without SMC3-LAP do not proliferate after deletion of *Smc3*.

F   Western blot showing HeLa cell extracts after treatment with siRNA as indicated and synchronized in G2-phase. Note that ESCO1 immunoblot signals were also reduced in sororin-depleted cells. We currently do not know if this is an effect of sororin depletion, an off-target effect or an artifact of unequal sample loading.

G   iFRAP quantification of stably chromatin-bound SMC3-LAP after siRNA treatment using cells synchronized in G2-phase. Error bars denote s.e.m., *n* > 11 cells per condition. Unpaired *t*-test was used to compare conditions.

stably associate with chromatin in G1-phase (Fig 2A) argue against the possibility that the stable association of these mutants with chromatin is a non-specific artifact, and further support the hypothesis that SMC3 acetylation promotes stable chromatin binding by enabling interactions with sororin.

## Sororin cannot stabilize cohesin–chromatin interactions before DNA replication

Sororin is present in S-phase and G2-phase, but not in G1-phase, when it is targeted for ubiquitin-dependent degradation by the APC/C bound to its co-activator Cdh1 (APC/C$^{Cdh1}$; Nishiyama *et al*, 2010; Rankin *et al*, 2005). The presence of sororin therefore correlates with the ability of cohesin to stably associate with chromatin in S-phase and G2-phase (Gerlich *et al*, 2006). In contrast, the presence of acetylated SMC3 does not correlate with the ability of cohesin to bind to chromatin stably, because in mammalian cells acetylated SMC3 also exists in G1-phase (Song *et al*, 2012; Whelan *et al*, 2012). We therefore tested if expression of sororin in G1-phase would enable stable binding of cohesin to chromatin.

For this purpose, we transiently expressed a sororin mutant which cannot be ubiquitinated by APC/C$^{Cdh1}$ and can therefore not be degraded, synchronized cells in G1-phase and measured

cohesin–chromatin interactions in iFRAP experiments. To generate a non-degradable version of sororin, we mutated three amino acid residues in the KEN box, a degron that is recognized by APC/C$^{Cdh1}$, to alanine residues (Pfleger & Kirschner, 2000; Rankin *et al*, 2005). Cells expressing wild-type SMC3-LAP or SMC3-LAP(QQ) were synchronized in early S-phase, were released into fresh medium, and transiently transfected with plasmid encoding the non-degradable sororin mutant tagged at its C-terminus with a Flag epitope (sororin$^{KBM}$-FLAG). In cells, which had progressed into G1-phase (Fig 3A), sororin$^{KBM}$-FLAG could be detected by immunofluorescence microscopy (Fig 3B; in 50% of cells) and by immunoblotting (Fig 3C). When analyzing the turnover of cohesin on chromatin in iFRAP experiments, all data could be fitted with a single exponential function, indicative of a single population of cohesin present in these cells (Appendix Fig S3). Also no significant change in residence time of wild-type SMC3-LAP or SMC3-LAP(QQ) was observed in cells which expressed sororin$^{KBM}$-FLAG (Fig 3D). The presence of sororin in G1-phase is therefore not sufficient to stabilize cohesin on chromatin, consistent with the observation that in *Xenopus* egg extracts SMC3 acetylation is not sufficient to recruit sororin to cohesin before DNA replication (Lafont *et al*, 2010; Nishiyama *et al*, 2010; Song *et al*, 2012).

We analyzed next if overexpression of the non-degradable sororin[KBM]-FLAG mutant could increase the number of stably chromatin-bound cohesin complexes in G2-phase, both to test the functionality of sororin[KBM]-FLAG, and to address if sororin levels may be limiting the number of stably chromatin-bound cohesin complexes in G2-phase. We therefore synchronized cells expressing wild-type SMC3-LAP or SMC3-LAP(QQ) in early S-phase and transiently transfected them with plasmid encoding sororin[KBM]-FLAG during the arrest. After releasing cells for 6 h into fresh medium, mock-transfected cells had progressed into G2-phase, whereas sororin[KBM]-FLAG trans-fected cells showed delayed cell cycle progression with the majority of cells still replicating DNA (Fig 3E). Sororin[KBM]-FLAG expression was detected in 25–30% of cells (Fig 3F). Under these experimental conditions, only 25% of cohesin complexes were stably chromatin bound (compared to 35–40% in previous experiments, see above) in mock-transfected cells, as measured in iFRAP experiments (Appendix Fig S3B and C). However, overexpression of sororin[KBM]-FLAG restored the number of stable interactions to more than 40% in cells either expressing wild-type SMC3-LAP or SMC3-LAP(QQ) (Fig 3G). As expected, depletion of ESCO1 and ESCO2 by RNAi abolished this effect in cells expressing wild-type SMC3-LAP, but not in cells expressing SMC3-LAP(QQ) (Fig 3G), indicating that sororin overexpression increased the number of stably chromatin-bound cohesin complexes by specifically interacting with complexes containing acetylated SMC3 or acetyl-bypass mutants of SMC3. Sororin could also increase stable cohesin–DNA interactions when its overexpression was induced exclusively in G2-phase by transfect-ing SMC3-LAP(wt) cells synchronized in G2-phase with sororin[KBM]-RFP in the presence of the CDK1 inhibitor RO-3306 (Fig EV1A–C). These results suggest that the levels of sororin in G2-phase are normally limiting the number of cohesin complexes that can stably associate with chromatin and further show that the sororin[KBM]-FLAG mutant used here is functional. The inability of sororin[KBM]-FLAG to stabilize cohesin in G1-phase is therefore not caused by a defect in its functionality but must instead be caused by a compo-nent or process that is absent in G1 but present in G2-phase.

In these experiments, we also tested under which conditions sororin[KBM]-FLAG can bind to cohesin. For this purpose, we released chromatin-bound proteins by benzonase treatment, immunoprecipi-tated sororin[KBM]-FLAG, and analyzed the presence of co-precipi-tating SMC3-LAP by immunoblotting. While we could detect a small amount of cohesin associated with overexpressed sororin in G1-phase, much more cohesin co-precipitated in G2-phase (Fig 3H). We did not see significant differences between cohesin complexes containing wild-type SMC3-LAP and SMC3-LAP(QQ) but noticed that acetylated forms of SMC3-LAP and endogenous SMC3 were enriched by co-immunoprecipitation with sororin[KBM]-FLAG (note that SMC3-LAP(QQ) cannot be recognized by these antibodies, Fig 2E). This is in agreement with the notion that sororin binds to cohesin on chromatin only if SMC3 is acetylated.

We had previously shown that the SMC3-LAP(QQ) and SMC3-LAP(RR) mutants efficiently co-immunoprecipitated sororin from soluble fractions of HeLa cells in G2-phase (Nishiyama *et al*, 2010). We speculated back then that SMC3 acetylation or acetyl-bypass mutations could lead to conformational rearrangements within cohesin that enable sororin binding. We assumed that this would normally only occur on chromatin, where SMC3 is acetylated, but that introduction of SMC3-LAP(QQ) or SMC3-LAP(RR) into cohesin

would allow these complexes to interact with sororin also in the nucleoplasm. However, our finding that sororin[KBM]-FLAG expressed in G1-phase does not efficiently associate with SMC3-LAP(QQ) on chromatin (Fig 3H) contradicts this idea. We therefore tested the ability of SMC3-LAP(QQ) to co-precipitate sororin[KBM]-FLAG from soluble fractions prepared in the above experiment. Consistent with our previous observations (Nishiyama *et al*, 2010), we found more sororin[KBM]-FLAG associated with cohesin containing SMC3-LAP (QQ) than cohesin containing wild-type SMC3-LAP (Fig EV1D). Unexpectedly, however, this was observed not only with samples from cells in G2-phase, but also when cells were synchronized in G1-phase. This means that soluble but not chromatin-bound cohesin containing SMC3-LAP(QQ) is able to bind sororin already before DNA replication. We do not know the reason for this difference but will speculate in the Discussion what it could mean.

## The genome-wide association of sororin with cohesin occurs exclusively on replicated DNA

The observation that the presence of acetylated SMC3 or acetyl-bypass mutants of SMC3 enables the recruitment of sororin to chro-matin-bound cohesin after but not before DNA replication (Lafont *et al*, 2010; Nishiyama *et al*, 2010; Song *et al*, 2012; Fig 3H) implies that DNA replication is also required for sororin–cohesin interac-tions on chromatin. The necessity for DNA replication to recruit sororin to cohesin could either reflect a global change in cell cycle signaling during S-phase, for example, an increase in S-phase protein kinase activities, or could result from local changes due to replication fork firing. Local changes could be the presence of a factor that is part of the replication fork or could be local alterations of cohesin, DNA, or chromatin. In order to distinguish between global and local effects of DNA replication on interactions between sororin and chromatin-bound cohesin, we analyzed the genome-wide distribution of sororin by chromatin immunoprecipitation followed by Solexa sequencing (ChIP-seq) in S-phase and G2-phase. We then compared the distribution of sororin to the appearance of replicated DNA in S-phase, which we analyzed by bromo-deoxyuridine (BrdU)-pulse DNA immunoprecipitation sequencing (DIP-seq). If the cell cycle state during DNA replication affected sororin binding globally, we expected to see a global increase of sororin–chromatin binding events from S-phase to G2-phase. If on the other hand DNA replication induced sororin binding to chro-matin locally, then sororin localization should follow the appear-ance of replicated DNA. Although this experiment could not measure sororin–cohesin interactions directly, we assume that the appearance of sororin on chromatin is a reflection of sororin–cohesin interactions, as the association of sororin with chromatin depends on cohesin (Lafont *et al*, 2010; Nishiyama *et al*, 2010). Consistent with this assumption, we found that almost all sites at which we could detect sororin in ChIP-seq experiments in either S- or G2-phase were also co-occupied by cohesin (see below).

This experiment is further based on the premise that in all cells in a given population the same genomic regions are synchronously replicated early in S-phase, whereas other regions are replicated later, which is thought to be the case (Cimbora & Groudine, 2001; White *et al*, 2004; Hiratani *et al*, 2008). We first tested if early and late replicating regions can be distinguished by DIP-seq in HeLa cells. For this purpose, we synchronized cells at the G1-S boundary by

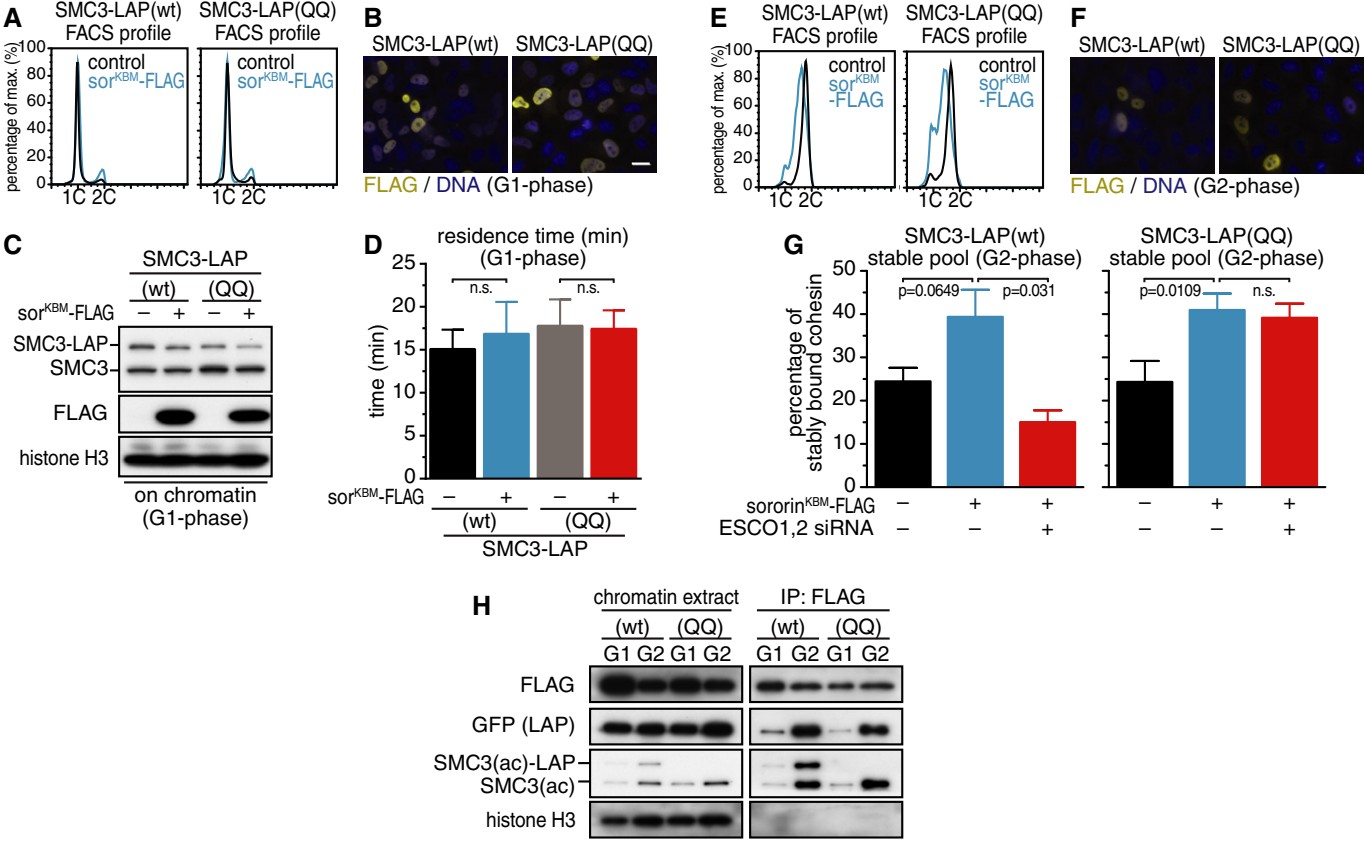

**Figure 3.  Sororin cannot stabilize cohesin–chromatin interactions before DNA replication.**

A    FACS profiles of cells transfected with non-degradable KEN box mutant sororin-FLAG (sor^KBM-FLAG) after synchronization in G1-phase and propidium iodide staining.

B    Immunofluorescence microscopy experiment to determine transfection efficiency. Scale bar, 10 μm.

C    Western blot of fractionated lysates to detect sororin^KBM-FLAG.

D    Quantification of SMC3-LAP residence time on chromatin after transfecting cells with sor^KBM-FLAG in G1-phase. Error bars denote s.e.m.; *n* > 15 cells per condition. Unpaired *t*-test was used to compare conditions.

E, F    Analysis of cell cycle distribution and transfection efficiency using cells synchronized in G2-phase.

G    Quantification of stably chromatin-bound SMC3-LAP from cells expressing non-degradable sororin in G2-phase. Cells were treated with siRNA as indicated. Error bars denote s.e.m.; unpaired *t*-test was used to compare conditions.

H    Western blot showing fractionated and nuclease digested chromatin from cells synchronized in G1- and G2-phase after immunoprecipitation with FLAG antibodies.

double thymidine arrest, released them for different periods of time, then incubated them with BrdU and analyzed the presence of regions into which BrdU had been incorporated by DIP-seq (Fig EV2A). Different regions of DNA were identified depending on when BrdU had been added, confirming that the same genomic regions were replicated at specific times during S-phase in most cells in the population (Fig EV2A and B). As expected, genomic regions into which BrdU was incorporated in early S-phase (ES in Fig EV2A and B) did not incorporate BrdU any longer in late S-phase (LS in Fig EV2A and B).

We next analyzed the genomic distribution of sororin in early S-phase by ChIP-seq and compared it to the distribution of cohesin, measured by SMC3-ChIP-seq, and to the distribution of replicated DNA, determined by DIP-seq. We also analyzed the distribution of sororin and SMC3 in G2-phase by using cells which had been released for 6 h from the G1-S arrest. SMC3 binding sites did not change significantly between early S-phase and G2-phase (35,757 sites in early S-phase, 37,691 sites in G2-phase, and 28,999 common

sites, as defined by MACS peak calling; Zhang *et al*, 2008b), consistent with the previous observation that cohesin occupies the same genomic sites in G1- and G2-phase (Wendt *et al*, 2008). In contrast, sororin was detected at fewer sites in early S-phase (8,300 sites) than in G2-phase (17,865 sites), as one would expect if sororin continues to be recruited to replicating regions during S-phase. As already mentioned, almost all of these overlapped with SMC3 sites (92.3% of sites in early S-phase, and 90.2% of sites in G2-phase). Importantly, manual inspection of the data revealed that the majority of sororin sites in early S-phase were found in regions which had been replicated at this time, as measured by DIP-seq (Fig 4A; dotted arrows). This was particularly evident when we binned the sororin ChIP-seq and DIP-seq data and compared their distribution side by side for individual chromosomes (Fig 4B).

To measure the correlation between sororin binding and BrdU incorporation in a quantitative and unbiased manner, we measured the significance of co-occurring sororin–BrdU peak regions using

interval statistics methods (IntervalStats; Chikina & Troyanskaya, 2012). Because the large BrdU containing regions could not be identified by the MACS peak calling algorithm, we identified these using the BayesPeak algorithm (Spyrou *et al*, 2009). For consistency, we re-analyzed sororin ChIP-seq data with the same algorithm, which identified 7,671 peaks in early S-phase and 14,811 peaks in G2-phase, and used these data for correlation with the DIP-seq data. Compared to randomized control simulations of the same datasets, this analysis indicated that sororin and BrdU incorporation co-occurred with high statistical significance in early S-phase (Fig 4C). The same result was obtained when we analyzed the distribution of sororin by ChIP-seq in cells synchronized in mid-S-phase (MS in Fig EV2A and B) and compared it to the corresponding DIP-seq profiles (Fig EV2C). A similar correlation was also observed in a second experiment in which we analyzed sororin binding and BrdU incorporation within two consecutive time "windows" in early S-phase (Fig EV2D–F).

However, once cells had progressed to G2-phase, most sororin peaks did not correlate with early replicating regions any longer (Fig 4C). This can partially be explained by the generation of additional sororin binding sites during DNA replication in middle and late S-phase (Fig EV2C). In addition, we observed that 49.6% of peaks identified by BayesPeak in early S-phase were not detected in our G2 dataset (i.e., only 3,869 of 7,671 peaks in early S-phase and 14,811 peaks in G2-phase were present in both data sets; according to the MACS algorithm, 34.7% of early S sororin peaks were not detectable in G2). As these results are based on four ChIP-seq experiments in early S-phase and two in G2-phase, they may reflect biologically relevant differences between sororin distribution in early S-phase and G2, rather than experimental variability. It is therefore possible that sororin is lost from some binding sites during progression through G2-phase and that, as a result, cohesion is also abrogated at these sites. In mechanistic terms, it is possible that this phenomenon is related to our observation that sororin turns over rapidly on chromatin-bound cohesin (see below).

SMC3 peaks both in early S-phase and in G2-phase also showed non-random co-occupancy with regions of early replication (Fig 4D), although to a much lesser degree than the sororin peaks found in early S-phase (Fig 4C). Because cohesin occupies the same genomic sites before and after DNA replication (Wendt *et al*, 2008), the co-occurrence of SMC3 peaks and early replicating regions could reflect a role of cohesin in spatially controlling DNA replication, as proposed by Guillou *et al* (2010), rather than a role of DNA replication in enabling recruitment of cohesin to specific sites in the genome.

Together, these data indicate that the ability of cohesin to recruit sororin is determined locally, and not globally. Local determinants of sororin recruitment could be the presence of the replication fork, the process of fork passage, the process of cohesion establishment, or a product of these processes. We performed further experiments to distinguish between these possibilities by using mouse cells in which the gene encoding sororin can be conditionally deleted. We will first describe this model before describing these experiments.

### The *Cdca5* gene encoding sororin is essential for development, cell proliferation, and proper cohesion

To be able to analyze the functions of sororin during embryonic development and in different cell types, we generated a conditional

sororin "knockout" mouse model by flanking exons 5 and 6 of the sororin-coding *Cdca5* gene with loxP sites (Fig 5A). Elimination of these exons is predicted to result in a premature stop codon, which prevents translation of almost 70% of the sororin polypeptide and thereby eliminates the conserved "sororin domain" (Nishiyama *et al*, 2010), which is required for cohesin binding and sister chromatid cohesion (Wu *et al*, 2011). Southern blotting showed correct targeting of the "floxed" allele (flx) and its deletion after crossing *Cdca5* flx/+ mice with mice expressing "MORE" Cre recombinase throughout the epiblast (Tallquist & Soriano, 2000) (Fig 5A). While mice heterozygous for the *Cdca5* deletion (*Cdca5* flx/Δ) were viable and appeared phenotypically normal, no mice carrying homozygous *Cdca5* deletions could be identified when analyzing newborn progeny of *Cdca5* flx/Δ crosses (Fig 5B). Also no embryos carrying homozygous *Cdca5* deletions could be recovered at E9.5 (Fig 5B), indicating that the *Cdca5* gene is already essential at early stages of development.

To analyze the role of sororin at the cellular level, we generated *Cdca5* flx/flx mice expressing a Cre-ERT2 transgene (Ruzankina *et al*, 2007) that can be activated using tamoxifen (4-OHT) and isolated mouse embryonic fibroblasts (MEFs; Fig 5C). When we deleted *Cdca5* from fibroblasts arrested in G0 and released these cells from quiescence by subculturing with serum addition, no sororin could be detected by immunoblotting (Fig 5D), whereas the levels of other cohesin proteins were unchanged compared to control cells. The cells lacking sororin were able to enter the cell cycle and to replicate DNA, as measured by appearance of cyclin A, incorporation of the thymidine analog 5-ethynyl-2′-deoxyuridine (EdU) into DNA during replication and by the appearance of Aurora B, which is expressed from S-phase onwards (Appendix Fig S4A and B). These results are consistent with previous analyses of cells in which sororin was depleted by RNAi and in which no defects in DNA replication had been observed (Rankin *et al*, 2005; Schmitz *et al*, 2007; Guillou *et al*, 2010; Lafont *et al*, 2010; Nishiyama *et al*, 2010).

In proliferation assays, *Cdca5*-deleted cells nevertheless increased in number much more slowly than cells containing one or two alleles of *Cdca5* (Fig 5E). Analyses of DNA content by propidium iodide staining and fluorescence activated cell sorting (FACS) revealed that many *Cdca5*-deleted cells became tetraploid or near-tetraploid within 5 days of release from quiescence, whereas the majority of control cells remained diploid (Fig 5F). Immunofluorescence microscopy showed that cells lacking *Cdca5* contained unaligned chromosomes more often than control cells within the first 2.5 days (Fig 5G and Appendix Fig S4C) and often contained multi-lobulated nuclei and micronuclei in interphase at later time points (Fig 5H).

These phenotypes are consistent with defects in chromosome segregation that can be caused by cohesion defects, which in turn are known to be caused by depletion of sororin (Rankin *et al*, 2005; Schmitz *et al*, 2007). Giemsa staining of spread chromosomes confirmed that cohesion defects existed in cells lacking sororin (Fig 5I and J). Unexpectedly, however, these phenotypes were less severe and penetrant than in HeLa cells from which sororin has been depleted by RNAi (Rankin *et al*, 2005; Schmitz *et al*, 2007). In *Cdca5*-deleted MEFs, cohesion was often lost only partially, resulting in sister chromatids which were separated at centromeres but still loosely connected (middle panel in Fig 5J). To test whether this difference is caused by different sororin depletion efficiencies, or by

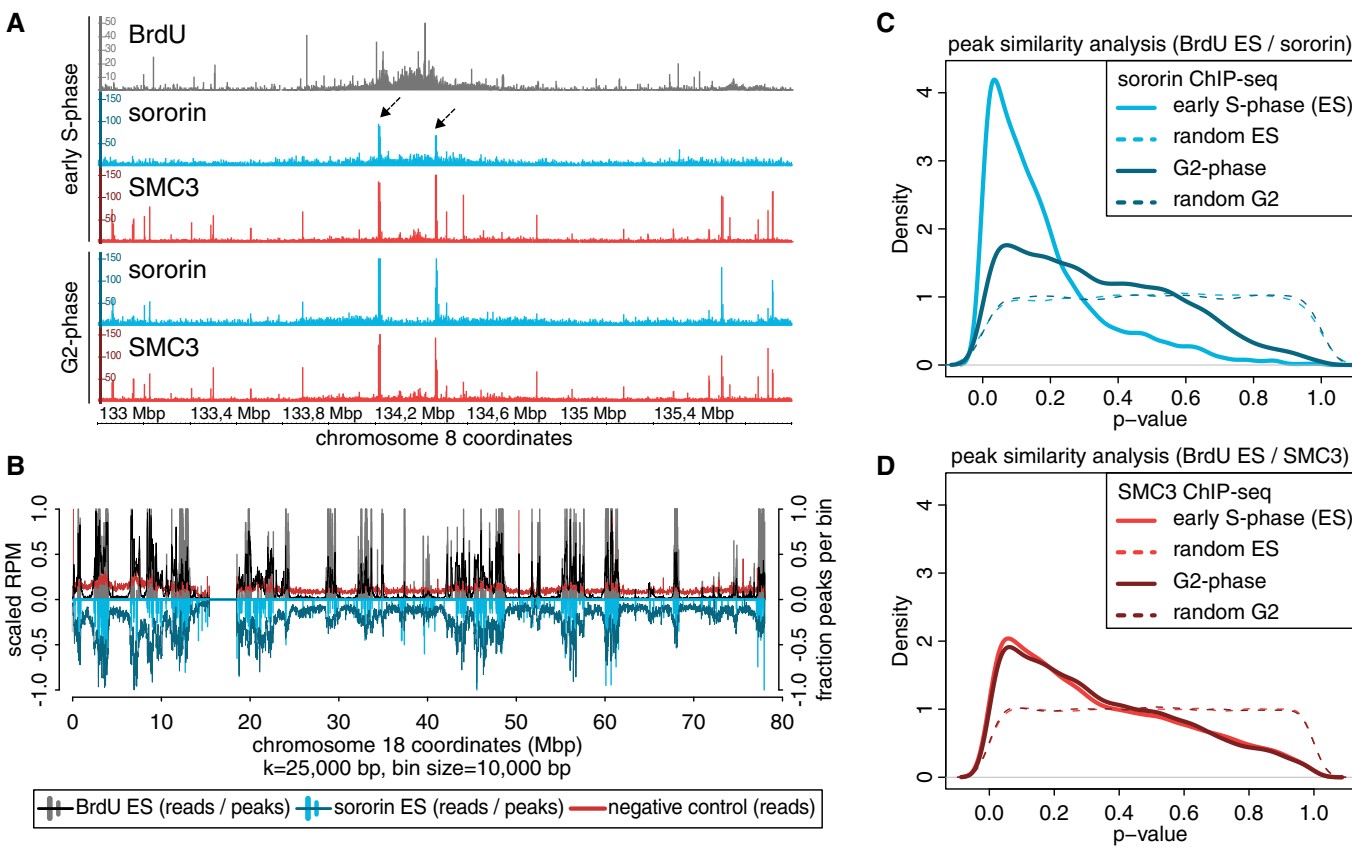

**Figure 4. The genome-wide association of sororin with cohesin occurs exclusively on replicated DNA.**

A   Examples of sororin (dotted arrows) and SMC3 ChIP-seq and BrdU DIP-seq data from early S-phase as compared to G2-phase using the Integrated Genome Browser (IGB; Nicol *et al*, 2009).

B   Alignment of sororin, BrdU, and BrdU-negative control sequencing bins in early S-phase to human chromosome 18 (GRCh37/hg19). Bars correspond to called peaks, curves to read enrichments after smoothing ($k$ = 25 kbp).

C   Quantification of the co-occupancy of BrdU incorporation and sororin localization in early S-phase and sororin in G2-phase depicted as *P*-value distributions calculated with IntervalStats (Chikina & Troyanskaya, 2012) and compared to randomized controls. Sororin peaks were identified as overlapping peaks of 4 samples in early S-phase and 2 samples in G2-phase, using only the common peaks.

D   Quantification of the co-occupancy of SMC3 in early S-phase and in G2-phase with BrdU incorporation in early S-phase.

differences between MEFs and HeLa cells, we also analyzed chromosomes from MEFs in which both alleles of *Smc3* were deleted by Cre-mediated recombination. In these cells, cohesion defects were slightly more penetrant than in *Cdca5*-deleted cells (Fig 5J), but not nearly as penetrant as the phenotype that is caused by cohesin depletion in HeLa cells (see Watrin *et al*, 2006 for an example). For reasons that are currently unknown, cohesion in MEFs is therefore less sensitive to reductions in cohesin levels than cohesion in HeLa cells. Nevertheless, our results confirm that sororin is required for proper cohesion in mouse cells, and they show that sororin is essential for early embryonic development.

**A *Cdca5* degron allele supports viability and sister chromatid cohesion in mouse fibroblasts**

To be able to analyze whether sororin is required for the maintenance of sister chromatid cohesion, we generated a version of sororin which can be inactivated at different times in the cell cycle. For this purpose, we modified a BAC containing a LAP-tagged

version of the mouse *Cdca5* locus (Whelan *et al*, 2012) by addition of an auxin-inducible degradation cassette (AID; Nishimura *et al*, 2009) to the 3′-end of the coding sequence (Fig 6A). We next generated MEFs with homozygous *Cdca5* deletion that stably integrated *Cdca5*-LAP-AID and a vector expressing the F-box transport inhibitor response-1 auxin receptor protein (TIR1; subsequently referred to as F-box protein) from *Oryza sativa* (Fig 6B). The resulting cells were viable and proliferated normally (unlike cells lacking sororin; Fig 5E), indicating that the sororin-LAP-AID protein is functional. We monitored sororin-LAP-AID signals by immunofluorescence microscopy using the GFP moiety of the LAP tag and found that after 24-h incubation of cells with auxin to induce degradation of sororin, the GFP fluorescence intensity became indistinguishable from "background" (Fig 6B and C, compare *Cdca5* +/+ cells with *Cdca5* Δ/Δ cells in the presence of sororin-LAP-AID, F-box, and auxin). When we analyzed GFP signal intensities of sororin-LAP-AID over time, we observed that GFP signals decreased within the first 6 h after auxin addition (Fig 6D). This decrease is slower than the decrease of some other AID-tagged proteins reported in the

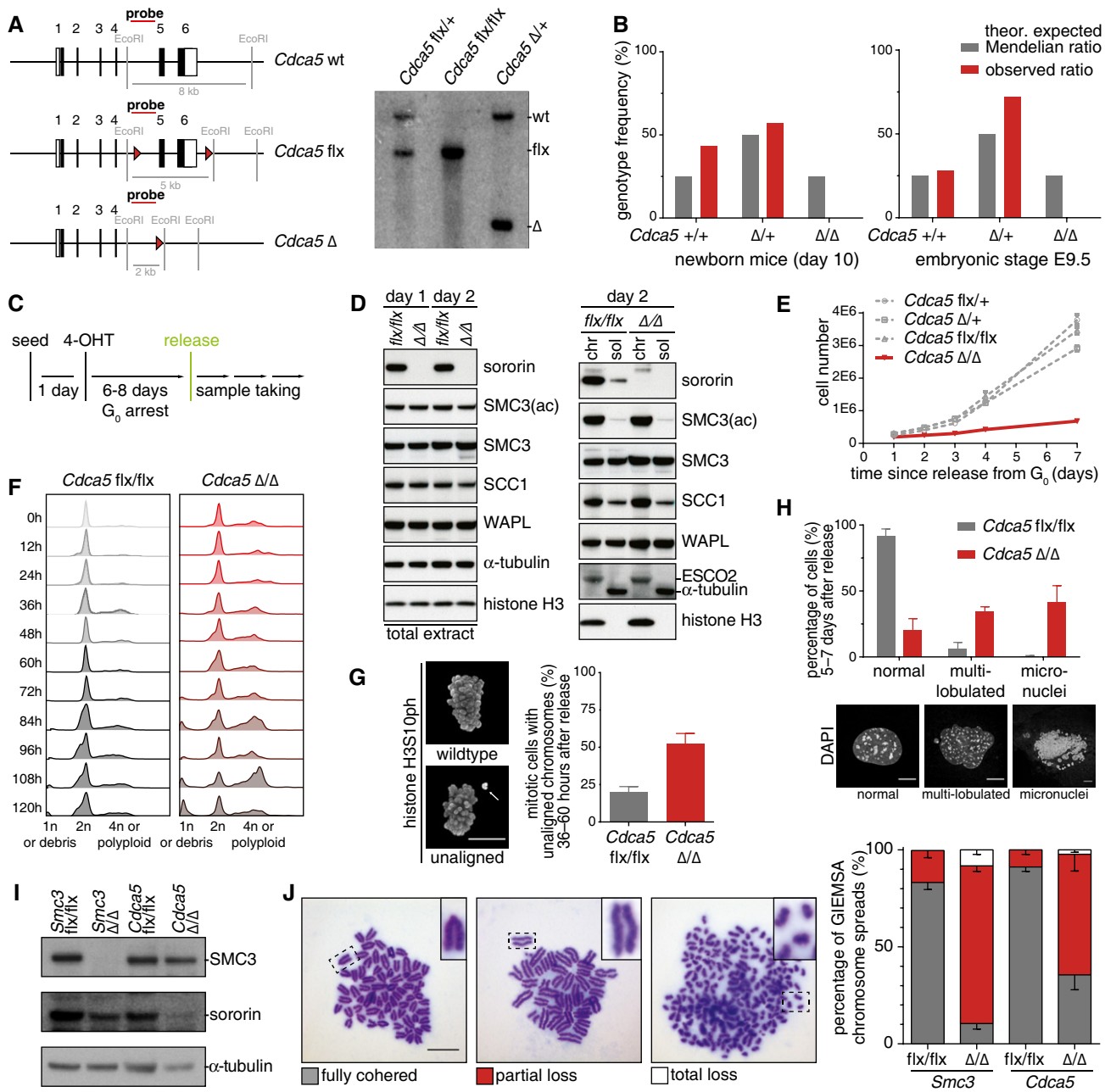

**Figure 5. The *CDCA5* gene encoding sororin is essential for development, cell proliferation, and proper cohesion.**

A   Schematic representation of wild-type (wt), targeted (flx), and deleted (Δ) *Cdca5* alleles (left) and Southern blot of mouse genomic DNA (right). Red triangles, loxP sites.

B   Observed and expected genotype distribution among offspring of heterozygous *Cdca5* Δ/+ mice (newborn mice, *n* = 136; E9.5, *n* = 54).

C   Gene deletion and G0-phase synchronization protocol for mouse embryonic fibroblasts isolated from *Cdca5* flx/flx mice expressing ERT2-Cre.

D   Western blot of cell extracts to analyze depletion of sororin protein from fibroblasts after tamoxifen-induced Cre expression. Extracts were prepared either one or two days after release from the arrest as indicated. Chr, fractionated chromatin; sol, soluble proteins.

E   Proliferation curve of cells with the indicated genotype after tamoxifen treatment.

F   Observed cell cycle distribution of fibroblasts treated with tamoxifen under starvation and subcultured in rich medium.

G   Quantification of mitotic cells with misaligned chromosomes (white arrow) after *Cdca5* deletion and proliferation for 36 to 60 h. Scale bar, 10 μm; error bars denote s.e.m.; *n* > 500 cells per condition.

H   Phenotypic classification of nuclear morphology after *Cdca5* deletion and proliferation for 5 to 7 days. Scale bar, 10 μm; error bars denote s.e.m.; *n* > 500 cells per condition.

I   Western blot showing protein depletion efficiency. Cre expression was induced with tamoxifen for 24 h in proliferating *Smc3* and *Cdca5* conditional knockout immortalized fibroblasts.

J   Analysis of chromosome spreads after *Smc3* or *Cdca5* deletion and nocodazole treatment. Representative images from every category are shown; scale bar, 10 μm; error bars denote s.e.m.; *n* > 500 per genotype.

literature (Holland *et al*, 2012), but still enabled us to test whether sororin's function is required to maintain cohesion once DNA replication has been completed (see below). To test whether the auxin-induced degradation system is able to deplete sororin-LAP-AID to the same degree as can be obtained by deletion of both *Cdca5* alleles by Cre recombinase, we compared cohesion defects in these cells. Giemsa staining of spread chromosomes from *Cdca5* Δ/Δ sororin-LAP-AID cells after 24-h incubation with auxin showed that cohesion was affected similarly as after *Cdca5* deletion and that this effect was dependent on the presence of auxin (Fig 6E). As expected, the presence of this phenotype correlated with the depletion of sororin-LAP-AID, as measured by immunoblotting (Fig 6F). The auxin degradation system thus enables time-resolved analysis of the function of sororin.

### Sororin, PDS5A, and WAPL turn over rapidly on chromatin

Because sororin can convert acetylated cohesin into a stably chromatin-bound form (Schmitz *et al*, 2007), we wanted to test whether sororin achieves this by itself stably associating with cohesin. We therefore performed photobleaching experiments on cells expressing GFP-tagged sororin. We used sororin-LAP-AID for this purpose, not because auxin-induced degradation was required to perform this experiment, but because our previous results had shown that this version of sororin is functional in the absence of endogenous sororin (Fig 6). We performed FRAP experiments (as opposed to iFRAP in the previous experiments) because FRAP can measure a different range of protein dynamics in the cell, from very dynamic to immobile, whereas iFRAP allows to specifically measure the recovery of slowly redistributing proteins.

We synchronized cells for 24 h by aphidicolin addition, released them into fresh medium for 2 and 7 h, and monitored cell cycle progression by propidium iodide staining (Fig 7A) and levels of sororin-LAP-AID by immunoblotting (Fig 7B). Sororin protein levels increased during S-phase when compared to asynchronously proliferating cultures (Fig 7B), presumably due to inactivation of APC/C$^{Cdh1}$ at the G1-S-transition. We then performed FRAP experiments by spot bleaching nuclear sororin-LAP-AID (Fig 7C) and analyzing its recovery over time (Fig 7D). Surprisingly, sororin-LAP-AID recovered already within a few minutes after photobleaching (Fig 7D), in contrast to photobleached SMC3-LAP that does not recover completely within more than 2 h in G2-phase (Fig 1). Despite small differences in the recovery slope, similar results were obtained for cell populations enriched in S-phase (aphidicolin and 2 h release) and in G2-phase (7 h release). Bi-exponential curve fitting indicated the presence of a small diffusive pool of sororin and of a much larger chromatin-bound pool, representing at steady state 72–84% of all nuclear sororin (Fig 7E). Recovery kinetics of the chromatin-bound pool corresponds to a residence time of 100 and 70 s in S-phase and G2-phase, respectively (Fig 7F). These results indicate that sororin stabilizes cohesin on chromatin by relatively dynamic association, that is, without itself becoming a permanent component of the cohesin complex it stabilizes.

In immunofluorescence microscopy experiments, we noticed that sororin-LAP-AID in MEFs changed in appearance from an overall nuclear localization in S-phase to densely stained regions in G2-phase that were reminiscent of heterochromatic foci (Fig 7C).

Photobleaching experiments confirmed that significantly more sororin-LAP-AID was present in heterochromatic foci than in other nuclear regions (84% at foci vs. 65.6% at non-foci; Appendix Fig S5A–D), but the residence of sororin-LAP-AID in heterochromatic foci was similar to the one measured in other regions (between 55 and 130 s). These data suggest that sororin–cohesin interactions are influenced via yet unknown mechanisms by the surrounding chromatin state. Since centromeric clusters at heterochromatic foci are the main sites of sister chromatid cohesion in early mitosis, it is possible that these regions are primed for cohesion maintenance already in late S/G2-phase.

Because the binding of sororin to cohesin depends on PDS5A and PDS5B, presumably because sororin directly binds to these proteins (Nishiyama *et al*, 2010), and sororin stabilizes cohesin on chromatin by inhibiting WAPL, we also analyzed how these proteins interact with chromatin. For this purpose, we generated HeLa cells in which sororin, WAPL, and PDS5A were N-terminally EGFP-tagged by CRISPR-mediated recombination, synchronized homozygously tagged clones in G2-phase and performed FRAP measurements (Fig EV3A–D). The measured residence time on chromatin for human sororin (100.5 s) was similar to mouse sororin-LAP-AID, although a lower fraction (52.3%) was chromatin bound. Importantly, EGFP-WAPL and EGFP-PDS5A turned over similarly quickly on chromatin as sororin, with mean residence times of 46.4 and 71.9 s, respectively (Fig EV3D). These results indicate that not only sororin but also the proteins that sororin interacts with and antagonizes are exchanging rapidly on cohesin which itself is bound to chromatin much more stably. Similar observations have been made for Wapl in budding yeast (Chan *et al*, 2012).

Because the addition of recombinant sororin to *Xenopus* egg extracts can increase the amount of WAPL on chromatin (fig S4F in Nishiyama *et al*, 2010), we tested if the dynamics of WAPL on chromatin could be modulated by sororin. For this purpose, we measured the behavior of EGFP-WAPL after depleting sororin by RNAi and synchronizing cells in G2-phase (Fig EV3E–I). However, quantification of the recovery kinetics compared to mock siRNA-treated cells showed no significant differences, indicating that sororin does either not influence WAPL dynamics on chromatin or only to an extent not detectable in our FRAP assay (Fig EV3I).

### Sororin is a cohesion maintenance factor

We next tested if sororin is required for cohesion maintenance. For this purpose, we used auxin-dependent degradation to inactivate sororin either during or after cohesion establishment in S-phase. We synchronized logarithmically growing *Cdca5* Δ/Δ sororin-LAP-AID MEFs at the beginning of S-phase by aphidicolin treatment (Fig 8A). We then released cells for 10 h with the addition of auxin at time-point 0 h (sororin degradation during S-phase) or 7 h (degradation in G2-phase), added nocodazole for 3 h (to arrest cells in prometaphase), and collected mitotic cells by shake-off from otherwise adherent cultures. To confirm that cells had been replicating DNA after time-point 0 and to exclude cells which had still been replicating DNA after time-point 7 h, we also supplemented medium with EdU while adding auxin (Fig 8A). The quantification of EdU incorporation showed indeed that the vast majority of cells replicated their DNA after release

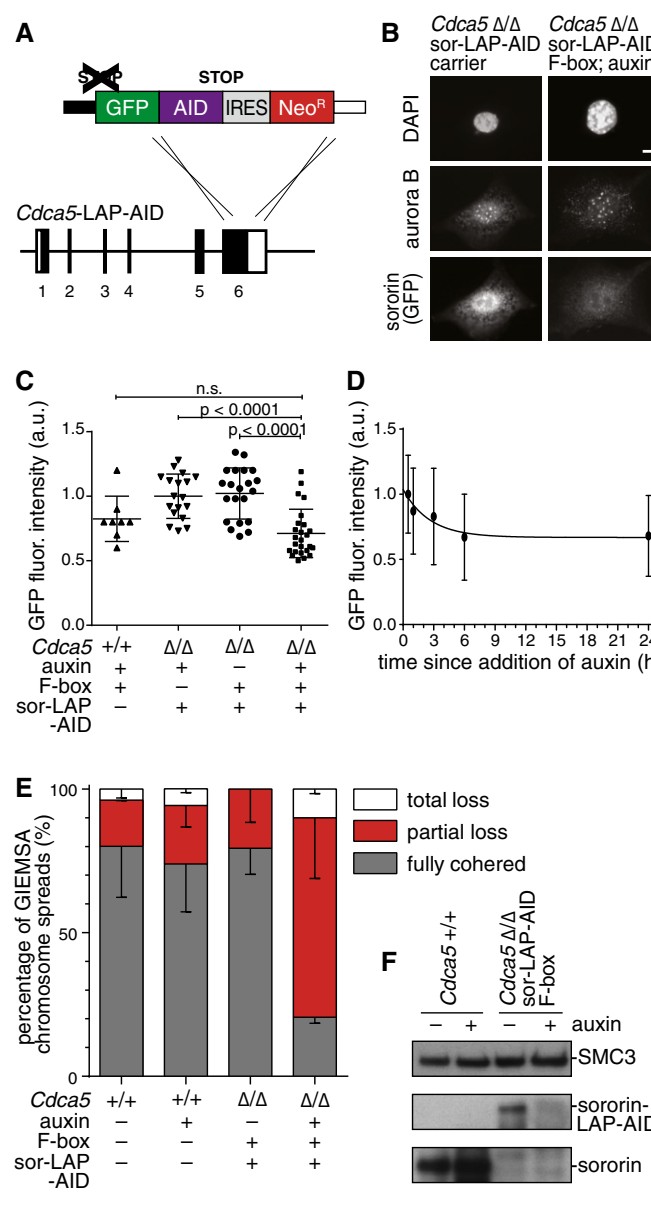

**Figure 6.  A sororin degron allele supports viability and sister chromatid cohesion in mouse fibroblasts.**

A   Schematic illustration of the LAP-AID-tag integrated into the bacterial artificial chromosome containing the mouse *Cdca5* locus.

B   Immunofluorescence staining of cells expressing sor-LAP-AID and the F-box helper protein after treatment with auxin. Antibodies against GFP mark the fusion protein and Aurora B antibody stains cells in G2/M-phase. Scale bar, 10 μm.

C   Quantification of immunofluorescence after treating cells with auxin. Error bars denote s.e.m.; Mann–Whitney *U*-test was used to compare conditions; a.u., arbitrary units.

D   Time-course quantification of GFP immunofluorescence signal decrease after auxin addition. Error bars denote s.e.m.; *n* > 16 per time point.

E   Analysis of chromosome spreads after auxin treatment. Error bars denote s.e.m.; *n* > 200 cells per condition.

F   Western blot showing cell extracts after treatment with auxin.

from the aphidicolin arrest (0 h) and that about 65% of cells had completed S-phase 7 h after the release (Fig 8B). Importantly, this was not dependent on the addition of auxin.

We then prepared chromosome spreads, stained them with Giemsa, and quantified their phenotypic appearance according to the classification in Fig 5J. The degradation of sororin during S-phase resulted in a large fraction of mitotic spreads showing partial cohesion defects (Fig 8C), similar to the data obtained after 24-h auxin incubation in asynchronous cultures (Fig 6E). Almost the same number of cells showed partial loss of cohesion when auxin was administered 7 h after releasing cells from the replication arrest (Fig 8C), although the majority of cells in this sample had completed DNA replication (Fig 8B) and therefore must have had established sister chromatid cohesion. To further test these assumptions, we also analyzed the co-occurrence of cohesion defects and EdU incorporation by staining spread chromosomes with DAPI (instead of Giemsa) and by simultaneous immunofluorescence labeling of EdU (Fig 8D). This allowed us to specifically identify cells that had been treated with auxin 7 h after aphidicolin release, which were EdU negative and which had therefore not been replicating their DNA any longer once sororin degradation was induced. Also in this experiment, cohesion defects were evident in the majority of cells, despite the fact that these had completed DNA replication by the time sororin was inactivated (Fig 8E). Finally, we tested if degradation of sororin could also destroy cohesion if auxin was added in mitosis (Fig EV4A–C). After enriching for mitotic cells by shake-off, we incubated sororin-LAP-AID cells in the presence of nocodazole and auxin and analyzed cohesion defects 6 h later. Also under these conditions, cohesion was impaired in the majority of cells following sororin degradation. These results indicate that the continuous presence of sororin is required to maintain sister chromatid cohesion in both G2-phase and prometaphase.

## Discussion

Sister chromatid cohesion is essential for chromosome bi-orientation and thus also for proper chromosome segregation. Knowledge about the mechanisms that lead to establishment and maintenance of cohesion is therefore essential for understanding how cells can segregate their chromosomes symmetrically. We have therefore tested here the hypothesis that in vertebrate cells cohesion is maintained by a multi-step mechanism, which depends on SMC3 acetylation and the subsequent binding of sororin (Lafont *et al*, 2010; Nishiyama *et al*, 2010). According to this hypothesis, sororin would maintain cohesion by inhibiting WAPL, which would otherwise release cohesin from DNA again and thereby precociously destroy cohesion (Nishiyama *et al*, 2010).

As predicted by this hypothesis, we show that the cohesin acetyltransferases ESCO1 and ESCO2 are required for sororin-dependent stabilization of cohesin on chromatin in G2-phase (Figs 1–3). By characterizing the behavior of SMC3 mutants that bypass the requirement for acetylation and thereby functionally (but not structurally) mimic the acetylated form of SMC3, we provide evidence that SMC3 acetylation is the only essential role of ESCO1 and ESCO2 in cohesin stabilization (Fig 2G). Somewhat unexpectedly, these experiments also revealed that MEFs which only express the acetyl-bypass mutants of SMC3 are viable (Fig 2E). This implies that cell cycle regulation of SMC3 acetylation, which is normally partially achieved by expression of ESCO2 specifically in S-phase (Hou & Zou, 2005; van der Lelij *et al*, 2009; Song *et al*, 2012; Whelan *et al*,

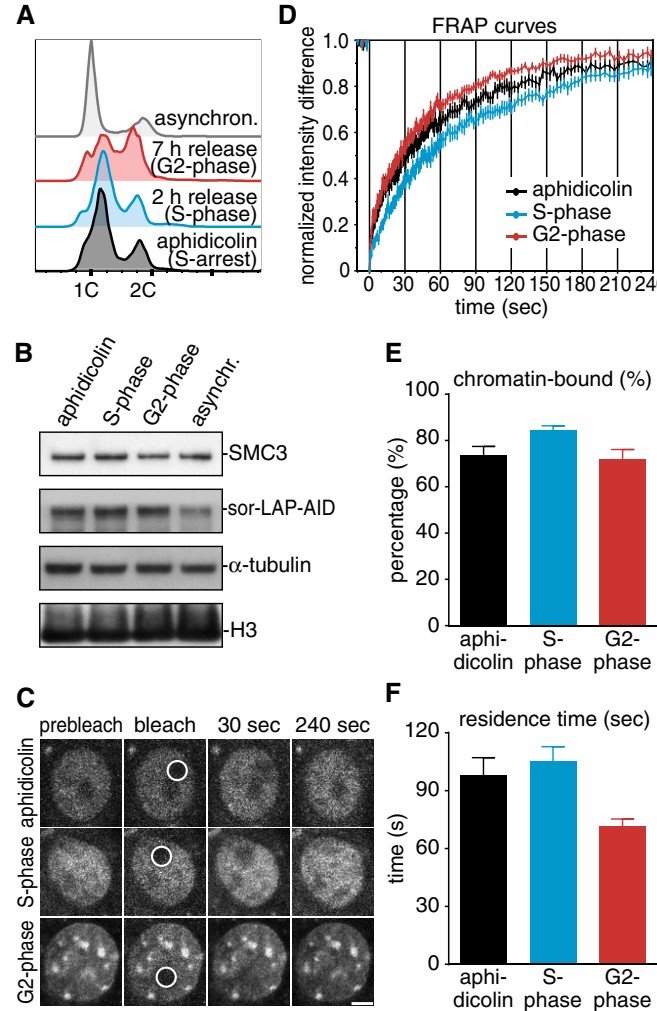

**Figure 7.  Sororin turns over rapidly on chromatin.**

A   Cell cycle distribution of immortalized *Cdca5* Δ/Δ sor-LAP-AID fibroblasts after 24-h incubation with aphidicolin and release.

B   Western blot showing cell extracts after incubation with aphidicolin and release.

C   Still images of a fluorescence recovery after photobleaching experiment. Scale bar, 5 μm. Bleach area radius, 2 μm.

D   Graph depicting the normalized intensity after photobleaching to quantify turnover of sororin (*n* = 15).

E   Quantification of the relative abundance of sororin-LAP-AID on chromatin. Error bars denote s.e.m.; *n* = 15.

F   Quantification of the chromatin residence time of sor-LAP-AID at different times. Error bars denote s.e.m.; *n* = 15.

experiments (Fig 3G) indicate that it is not the number of acetylated cohesin complexes but instead the levels of sororin that are limiting for how many cohesin complexes remain stably bound to chromatin throughout G2-phase. In future, it will be interesting to test why sororin levels are limiting and as a result only some cohesin complexes become stably associated with chromatin. For example, it is conceivable that cells limit the number of stably chromatin-bound cohesin complexes because a certain number of cohesin complexes that interact with chromatin dynamically is needed for other functions, such as mediating higher-order chromatin structure or gene regulation.

Because acetylated SMC3 exists throughout the cell cycle, but cohesin complexes that are stably bound to chromatin can only be detected in S- and G2-phase (Gerlich *et al*, 2006), SMC3 acetylation cannot be sufficient for stabilization of cohesin on chromatin. Our work here as well as previous observations (Gerlich *et al*, 2006; Schmitz *et al*, 2007) indicate that both sororin and DNA replication are also required for the formation of stable cohesin–DNA interactions. What could the role of DNA replication in this process be? It would have been plausible to think that a component of the replisome is required to recruit sororin to cohesin, similar to how in budding yeast Eco1/Ctf7, the ortholog of ESCO1 and ESCO2, is thought to be recruited to DNA by replication forks (Lengronne *et al*, 2006; Moldovan *et al*, 2006). Unexpectedly, however, two of our results argue against this possibility. One is our previous observation that sororin can also bind to soluble cohesin if it contains acetyl-bypass mutants of SMC3 (Nishiyama *et al*, 2010), implying that sororin can bind to these complexes in the absence of replisome components. The other finding is the surprising result that sororin can stabilize cohesin on chromatin while itself interacting with cohesin dynamically, with an average residence time on chromatin in the range of only a minute (Fig 7). This implies that sororin must be able to dissociate from cohesin and to re-bind to it frequently during G2-phase, that is, at a time where no replisomes or replication forks exist. Instead, it must be some property of cohesin itself that allows sororin to bind. This property cannot be the presence of acetylated SMC3 alone, because this also exists in G1-phase where sororin cannot bind to cohesin on DNA.

Instead, we initially suspected it must be some property that cohesin can only acquire when it is associated with replicated DNA. Curiously, however, we found that sororin can also bind to cohesin containing SMC3 acetylation bypass mutants in G1-phase if these complexes are not associated with DNA, as if cohesin can exist in two different states or conformations. One of these would be competent for sororin binding and would exist both in solution and on replicated DNA, and another one, which cannot interact with sororin, would only exist when cohesin is bound to unreplicated DNA. Normally, sororin binding is of course only observed for cohesin complexes that are acetylated and bound to replicated DNA, but not for soluble cohesin complexes because these are not acetylated. As unexpected as this may seem, the mutation of SMC3's acetylation sites may have revealed that there are important differences between soluble and DNA-associated cohesin complexes in G1-phase. Future work will be required to elucidate what these differences could be.

Our generation of a version of sororin which can be degraded upon addition of auxin enabled us to test if sororin is still required for cohesion once DNA replication has been completed. We found

2012) and partially by deacetylation of SMC3 by HDAC8 (Deardorff *et al*, 2012), is dispensable for viability, at least in fibroblasts. However, this finding is consistent with the observation that in mammalian cells SMC3 acetylation is observed throughout the cell cycle, presumably because ESCO1 is constantly present in these cells (Song *et al*, 2012; Whelan *et al*, 2012) and associates with cohesin binding sites throughout interphase (Minamino *et al*, 2015; Rahman *et al*, 2015). These results imply that the timing of SMC3 acetylation does not have to be strictly regulated in mammalian cells. Furthermore, the results of our sororin overexpression

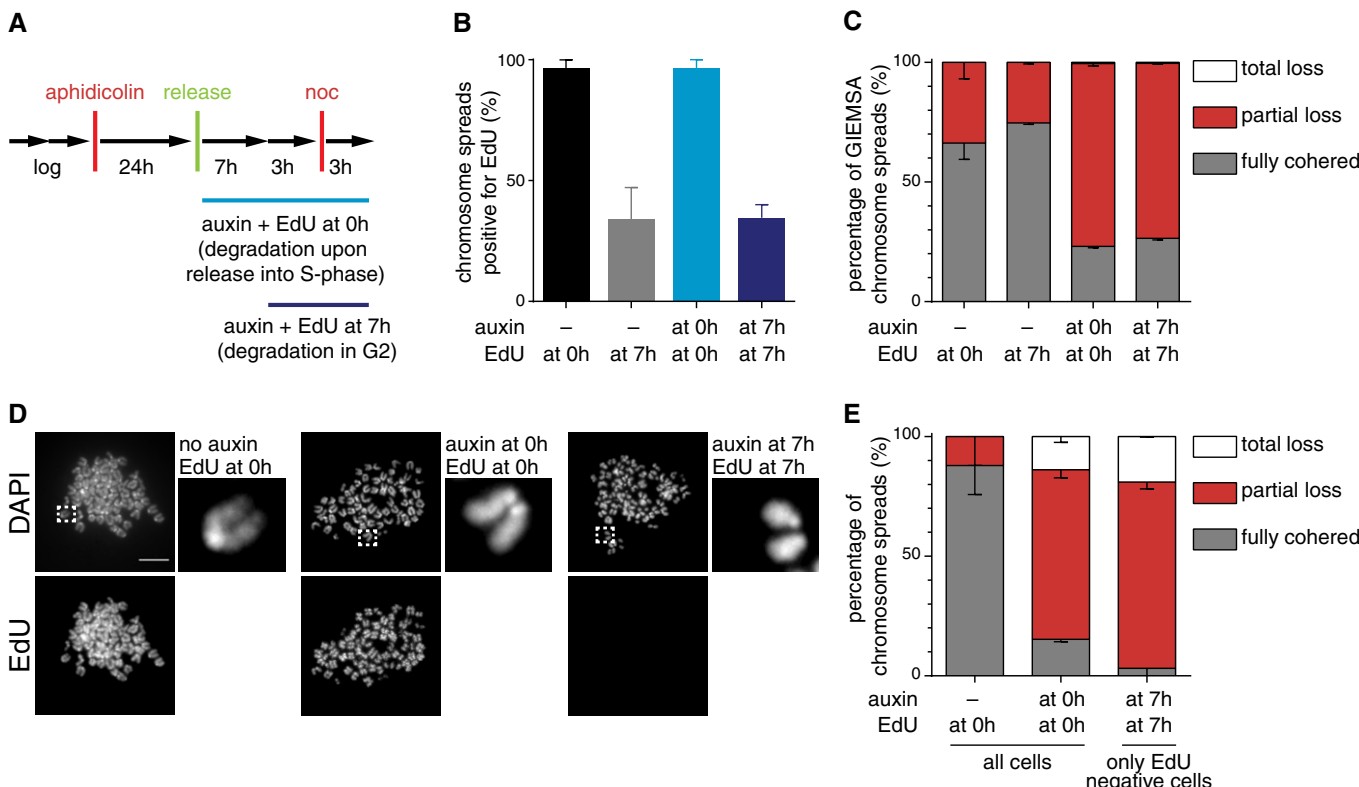

**Figure 8.  Sororin is a cohesion maintenance factor.**

A   Experimental setup used for cell synchronization, auxin, and EdU treatment of *Cdca5* Δ/Δ sor-LAP-AID F-box mouse embryonic fibroblasts.
B   Quantification of EdU-labeled prometaphase cells after treating cells with auxin and EdU at the indicated time points. Error bars denote s.e.m.; *n* > 200 cells per condition.
C   Analysis of chromosome spreads after auxin and EdU treatment and Giemsa staining. Error bars denote s.e.m.; *n* > 200 cells per condition.
D   Representative picture of the most prominent phenotype class upon auxin and EdU treatment and immunofluorescence microscopy. Scale bar, 10 μm.
E   Analysis of chromosome spreads after auxin treatment and EdU and DAPI labeling. Error bars denote s.e.m.; *n* > 200 cells per condition.

that this is clearly the case, that is, that sororin is required for the maintenance of cohesion. Our results do not exclude the possibility that sororin is also required for establishment of cohesion during DNA replication. However, we suspect that this is not the case because cells can establish cohesion normally in the absence of sororin, provided that WAPL has been depleted (Nishiyama *et al*, 2010). Taken together, the previous observation that sororin recruitment to cohesin depends on SMC3 acetylation (Lafont *et al*, 2010; Nishiyama *et al*, 2010) and our finding here that sororin is required for maintenance of cohesion seem to contradict the notion that in budding yeast Eco1/Ctf7 is only required during S-phase, that is, presumably only during the establishment of cohesion (thus its name "establishment of cohesion 1"; Skibbens *et al*, 1999; Toth *et al*, 1999). However, this conundrum could be explained if in budding yeast SMC3 acetylation generated in S-phase persisted long enough to maintain cohesion throughout G2-phase. Observations which imply that acetylation may persist even after inactivation of Eco1/Ctf7 have indeed been reported in budding yeast (Beckouet *et al*, 2010).

In summary, our results support the view that an essential function of SMC3 acetylation by ESCO1 and ESCO2 is to enable recruitment of sororin and thereby to maintain cohesion from DNA

replication until mitosis by inhibition of WAPL. We do not know yet if this is the only role of SMC3 acetylation in cohesion. To test this, it will be interesting to delete the genes encoding ESCO1 and ESCO2 in cells expressing acetyl-bypass mutants of SMC3. Likewise, it will be interesting to understand if similar mechanisms of cohesion maintenance exist in eukaryotes other than vertebrates. Genetic observations in fungi and plants imply that this is the case as orthologs of ESCO1 and ESCO2 are only essential for viability in these organisms in the presence of WAPL, but not in its absence (Tanaka *et al*, 2001; Ben-Shahar *et al*, 2008; Bernard *et al*, 2008; Feytout *et al*, 2011; Chan *et al*, 2012; Lopez-Serra *et al*, 2013; De *et al*, 2014), reminiscent of our finding that sororin is only essential for cohesion in the presence of WAPL (Nishiyama *et al*, 2010). However, there may also be important differences between the mechanisms of cohesion maintenance between metazoans and other eukaryotes as orthologs of sororin have so far only been discovered in vertebrates and insects (Rankin *et al*, 2005; Nishiyama *et al*, 2010) but not in fungi and plants. Furthermore, based on genetic observations in budding yeast, it has been argued that Eco1/Ctf7 may have cohesion functions other than antagonizing Wapl (Guacci *et al*, 2015).

Molecular knowledge about how cohesion is maintained may be particularly important for understanding how bivalent chromosomes

are held together for many years in human oocytes. In these cells, cohesion can only be established during pre-meiotic DNA replication (Tachibana-Konwalski *et al*, 2010), similar to the situation in somatic cells. This occurs before birth and results in cohesion which has to be maintained until meiosis is completed. This occurs only during follicular maturation cycles after puberty, that is, in humans many years later. Gradual loss of cohesion during these long periods of time is thought to be a major cause of the maternal age effect of oocyte aneuploidy, which can result in trisomy 21 and spontaneous abortions (Hunt & Hassold, 2010). The conditional *Cdca5* (*Sororin*) "knockout" mouse model, which we have characterized here, and another recently established model which enables inactivation of *Wapl* (Tedeschi *et al*, 2013) may be useful to study how cohesion can be maintained for such long periods of time in mammalian oocytes.

# Materials and Methods

## Antibodies

Polyclonal rabbit antiserum was raised against human sororin for ChIP (A954; C-MNAEFEAAEQFDLLVE) and guinea pig antiserum against mouse sororin (A1031; recombinant full-length sororin). Polyclonal antibodies against SMC3 (A845; Sumara *et al*, 2002), ESCO1 and ESCO2 (A782; A784; Nishiyama *et al*, 2010), sororin (A953; Schmitz *et al*, 2007), SCC4 (A974; Watrin *et al*, 2006), and WAPL (A960; Tedeschi *et al*, 2013) have been described previously. Mouse antibody to acetyl-SMC3 was a gift from K. Shirahige, and guinea pig antibody against ESCO2 (Fig 5D) was a gift from J. de Winter. The following commercial antibodies were used: α-tubulin (Sigma-Aldrich, T5168), histone H3 (Cell Signaling, 9715L), FLAG (Sigma-Aldrich, F3165), GFP (Roche, 11814460001), BrdU (BD Biosciences, 347580), SMC3 (Bethyl Laboratories, A300-060A), SMC1 (Bethyl Laboratories, A300-055A), SCC1 (Millipore, 05-908) antibodies, and rabbit IgG (Invitrogen, 10500C).

## Generation of conditional *Cdca5* allele and mice

A targeting vector (Nakashima *et al*, 2011) for conditional deletion of the *Cdca5* gene was amplified by PCR from BAC bMQ-410a11 containing the *Cdca5* gene of the 129/Sv mouse line to include exons 5 and 6 and 4.5 kb upstream and 1.5 kb downstream. HM-1 mouse embryonic stem cells were used for gene targeting. After electroporation and selection, single colonies were picked into 96-well plates and analyzed by long-range PCR for correct integration of targeting vector before Southern blot analysis (Madisen *et al*, 2010) using a probe amplified with primers CAGTACCTTAGCCT CAAGTG and ACTGCAGGCGAAGCTAGAAC as outlined in Fig 5. Correctly targeted clones were expanded and injected into C57BL/6 blastocysts.

## Mouse crossing

Chimeric mice were crossed to FLPe expressing mice (Rodriguez *et al*, 2000) in order to remove the FRT neomycin resistance cassette. Subsequent generations were backcrossed to wild-type C57BL/6 mice. The flox allele was converted into a delta allele by

crossing with MORE mice expressing Cre recombinase (Tallquist & Soriano, 2000), followed by crossing male double positive offspring with wild-type mice. Conditional knockout mice for *Smc3* were acquired from EUCOMM and will be described elsewhere. Tamoxifen inducible Cre recombinase was introduced by crossing with Cre-ERT2 transgenic mice (Ruzankina *et al*, 2007). All animal experiments were carried out according to valid project licenses, which were approved and regularly controlled by the Austrian Veterinary Authorities.

## Mouse embryonic fibroblast cultures, immortalization, and synchronization

Mouse embryonic fibroblasts (pMEFs) were isolated from E13.5–14.5 (*post-coitum*) embryos as described previously (Michalska, 2007). Cells were cultivated in pMEF medium (15% (v/v) FBS; 0.2 mM L-glutamine; 100 U/ml penicillin and 100 μg/ml streptomycin; 1 mM sodium pyruvate; 100 μM non-essential amino acids; 100 μM β-mercaptoethanol). For immortalization, primary embryonic fibroblasts were treated with retroviruses expressing shRNA against p53 (Dow *et al*, 2012). For synchronization, fibroblasts were grown to confluency and treated with 0.5 μM 4-hydroxytamoxifen (4-OHT; Sigma) or carrier for 3–5 days. Medium was changed every 2 days. Cells were then released into the cell cycle by splitting 1:5 into medium containing 4-OHT or carrier. For analysis of cell cycle progression, EdU (Life Technologies) was added to the medium to 10 μM final concentration for 30 min before fixation.

## LAP-AID-tag and BAC recombineering

Smc3-LAP BAC constructs were introduced using Fugene HD transfection reagents. Cells were then selected based on geneticin (G418) resistance (0.33 mg/ml final concentration) and thereafter FACS sorted based on GFP expression levels. Cells were then cultured for 2 days in the presence of tamoxifen to delete the endogenous copy of Smc3. After two more passages, Western blot samples were taken and the remaining cells were further kept in culture. The LAP-AID tag was created by inserting the auxin-inducible degron (AID) sequence (Nishimura *et al*, 2009) between the GFP and the IRES: gb3:neo cassette described in Poser *et al* (2008). For generation of *Cdca5*-LAP-AID, the tag was inserted at the C-terminus of mouse sororin by recombination into the BAC RP23-477H14. The modified BAC was transfected into immortalized fibroblasts using Fugene6 (Promega).

## Auxin degradation

Cells expressing sororin-LAP-AID were stably transfected with a vector containing the F-box transport inhibitor response 1 (TIR1) protein under control of a doxycycline-inducible promoter. To induce degradation, cells were treated with 1 μg/ml doxycycline (Sigma) and 500 μM indole-3-acetic acid (IAA; Sigma) for 3–24 h.

## Chromosome spreads

Cells in log phase or released for 7 h from aphidicolin arrest (1 μg/ml final concentration) were treated with 330 nM nocodazole for

3–5 h before mitotic shake-off, hypotonic treatment, spreading, and Giemsa staining. For analysis of chromosome morphology combined with EdU incorporation assay, hypotonically treated cells were spun on glass slides, fixed with paraformaldehyde, and stained for EdU incorporation using the Click-iT EdU Alexa Fluor 647 Imaging Kit (Life technologies) followed by counterstaining with DAPI.

## Immunofluorescence microscopy

Cells grown on coverslips were PBS-washed and fixed with 4% paraformaldehyde in PBS. After fixation, cells were permeabilized with 0.1% Triton X-100 in PBS for 5 min, blocked with 3% BSA in PBS containing 0.01% Triton X-100, and incubated with primary and secondary antibodies (Molecular Probes). DNA was counterstained with DAPI. Coverslips were mounted onto slides with ProLong Gold (Molecular Probes). Images were taken on a Zeiss Axioplan 2 microscope with 63× Plan-Apochromat objective lens (Zeiss). The system was equipped with a CoolSnapHQ CCD camera (Photometrics).

## HeLa cell culture, FACS analysis, RNAi, and plasmid transfection

HeLa Kyoto cells with and without a bacterial artificial chromosome encoding mouse Smc3-LAP wild-type, K105Q/K106Q, or K105R/K106R mutant genes were cultured as described previously (Nishiyama *et al*, 2010). Cells were synchronized at the G1/S-phase boundary by two consecutive arrest phases with 2 mM thymidine and released into fresh medium for 6 h (G2-phase), 15 h (G1-phase), or as indicated above for different states of S-phase. Cell cycle profiling was performed using propidium iodide staining (Ladurner *et al*, 2014). Transfection with siRNA was performed as described (van der Lelij *et al*, 2014). Plasmid encoding FLAG- or RFP-tagged sororin KEN box to alanine mutant under CMV promoter was generated by PCR. Plasmids were pre-mixed with Lipofectamine 2000 (Invitrogen) according to the manufacturer's instructions and added to cells at a final concentration of 0.9 μg/ml at 14–16 h before iFRAP analysis.

## HeLa Kyoto CRISPR-mediated genome engineering

HeLa Kyoto cells expressing N-terminally EGFP-tagged sororin, WAPL, and PDS5A proteins were generated by CRISPR-mediated homologous recombination as described (Cong *et al*, 2013). Cells were transfected with 2 plasmids expressing SpCas9(D10A) nickase and chimeric guide RNAs as well as a homology plasmid that carried the EGFP coding sequence flanked on either side by 500–1,000 nucleotides surrounding the target gene start codon that were generated from HeLa genomic DNA by PCR. Clonal cell lines were generated by FACS and recombination, and homozygous tagging was assayed by PCR and immunoblotting.

## Photobleaching microscopy

The inverse fluorescence after photobleaching setup has been described previously (van der Lelij *et al*, 2014). For photobleaching of mouse sororin-LAP-AID *Cdca5* Δ/Δ, cells were imaged on a Zeiss LSM5 Duo confocal microscope using a 63× Plan-Apochromat objective. Ten pre-bleach images were acquired before bleaching a radial

spot (*r* = 2 μm) three times at 100% laser intensity (100 mW diode 488) and acquiring 240 images at one-second intervals. Photobleaching of N-terminally tagged HeLa sororin, WAPL, and PDS5A cells was performed at the Stanford University School of Medicine Department of Biochemistry (USA) using a Nikon eclipse Ti microscope with an Apo TIRF 100× objective and equipped with an Andor iXon X3 camera and mosaic for spot bleaching, purchased using funds from a NIH S10 Shared Instrumentation Grant (S10RR026775-01). Signal intensities were measured using ImageJ at bleached, nuclear and background regions, and normalized according to Ellenberg *et al* (1997). Data were analyzed using Berkeley Madonna software and curve fitting ($a = (1-dS)*(1-EXP(-(kOff1)*time)) + dS*(1-EXP(-(kOff2)*time)))$).

## HeLa cell extracts, immunoblotting, and immunoprecipitation

Cell pellets were resuspended in extraction buffer (25 mM Tris pH 7.5, 100 mM NaCl, 5 mM MgCl$_2$, 0.2% NP-40, 10% glycerol, 1 mM NaF, 10 mM sodium butyrate, complete protease inhibitor mix (Roche)) and lysed on ice by passing through a hypodermic needle. To produce total extracts, a fraction of homogenate was incubated with DNase I (2 U/μl) for 60 min at 4°C and denatured in Laemmli's sample buffer. To separate soluble and chromatin-bound proteins, the homogenate was spun at 1,300 *g* and the chromatin pellet was washed 3 times with extraction buffer. Pellets were resuspended in Laemmli's sample buffer, heated to 95°C, and passed over a 0.45-μm filter. Immunoblotting was performed as described (Watrin *et al*, 2006). To release proteins from chromatin, samples were treated with benzonase (250 U/ml) and insoluble material removed by centrifugation. Extract was added to cross-linked antibody beads, incubated, washed, and eluted with 0.1 M glycine pH 2.

## Chromatin and BrdU-DNA immunoprecipitation

Cells were synchronized by double thymidine arrest and released into S-phase. Following a pulse of 50 μM BrdU, samples were processed for cross-linking, lysis, sonication, reverse cross-linking, and fragment size examination as described for chromatin immunoprecipitation in ChIP followed by sequencing. For DNA-IP, 300 μg DNA per condition was reverse cross-linked, purified, and ethanol precipitated. Illumina sample preparation was performed as recommended by the manufacturer (NEB; E6040). In brief, purified, sheared genomic DNA was blunt-end repaired, followed by addition of dA tails and sequencing adaptor ligation. After each step, DNA was purified using MinElute PCR purification kit (Qiagen). Finally, 500 ng of BrdU-labeled or non-labeled DNA as a control in 184 μl TE buffer was heat-denatured for 10 min and transferred to an ice-water bath for 2 min. After addition of 24 μl 10× IP buffer (100 mM sodium phosphate pH 7.2; 1.4 M NaCl; 0.5% Triton X-100) and 1 μg anti-BrdU antibody, samples were rotated for 2.5 h at 4°C. Fifteen μl protein-G dyna-beads (Invitrogen; 10004D) were pre-incubated with 500 μl 0.1% BSA in PBS for 2 h, resuspended in 30 μl IP buffer, added to samples, and incubated for 2 h. Samples were washed 3 times for 10 min with 350 μl IP buffer at room temperature, digested in 125 μl proteinase K buffer (50 mM Tris pH 8.0; 10 mM EDTA; 0.5% SDS; 35 μg proteinase K), and incubated at 55°C for 30 min

with shaking. Supernatant was purified with QIAquick MinElute columns in 17 μl elution buffer. DNA was PCR amplified for 18 cycles with Illumina adaptor primers and Phusion polymerase as described in the manufacturer protocol and purified via Qiagen PCR purification kit.

### Chromatin immunoprecipitation (ChIP) followed by sequencing

ChIP was performed as described (Wendt *et al*, 2008). Cross-linking was done with 1% formaldehyde for 10 min and quenched with 125 mM glycine. Cells were washed with PBS, lysed with 750 μl lysis buffer (50 mM Tris–HCl pH 8.0, 10 mM EDTA pH 8.0, 1% SDS, protease inhibitors (Roche; 11 873 580 001)) per 145-mm dish, scraped off the plate, and sonicated to fragment size peaking between 200 and 400 basepairs. Affiprep Protein A beads were pre-coated by an overnight rotation at 4°C in buffer Wash1 (20 mM Tris–HCl pH 8.0, 2 mM EDTA pH 8.0, 1% Triton X-100, 150 mM NaCl, 0.1% SDS, 1 mM PMSF) supplied with 0.1 mg/ml BSA and 0.5 mg/ml sheared salmon sperm DNA (10 bead volumes). A total of 1.5–2 mg DNA per IP was used. Samples were diluted 5 times with buffer Wash1 and pre-cleared with 100 μl pre-absorbed beads with rotation for 1 h at 4°C. Overnight immunoprecipitation with sororin antibody or non-immune rabbit IgG was followed by 2-h incubation with 100 μl pre-coated beads. Beads were washed twice for 10 min with buffer Wash1, twice for 3–5 min with buffer Wash2 (20 mM Tris–HCl pH 8.0, 2 mM EDTA pH 8.0, 1% Triton X-100, 500 mM NaCl, 0.1% SDS, 1 mM PMSF), twice with buffer Wash3 (10 mM Tris–HCl pH 8.0, 2 mM EDTA pH 8.0, 250 mM LiCl, 0.5% NP-40, 0.5% deoxycholate), twice with TE buffer, and eluted twice with 200 μl elution buffer (25 mM Tris–HCl pH 7.5, 5 mM EDTA pH 8.0, 0.5% SDS) for 20 min by shaking at 65°C. The eluates were treated with 250 μg RNase-A and 250 μg proteinase K at 37°C for 1 h and at 65°C overnight to reverse chemical cross-links. Then, addition of 1 μl glycogen (20 mg/ml) and 1/10 vol. sodium acetate (3 M, pH 5.2) was followed by extractions with phenol/chloroform/isoamylalcohol (25:24:1) and chloroform. DNA was precipitated with ethanol and dissolved in dH$_2$O.

Sequencing was carried out using the Illumina Genome Analyzer II system according to the manufacturer's protocol. Sequencing data discussed in this paper are available at the European nucleotide archive (ENA) under accession number PRJEB12214 (http://www.ebi.ac.uk/ena/data/view/PRJEB12214).

### Read mapping, peak calling, and overlap analysis

Sequencing reads were mapped with bowtie version 0.12.5 (Langmead *et al*, 2009) against human genome assembly GRCh37/hg19, allowing for two mismatches and outputting only uniquely aligned reads (parameters: -v 2 –best –strata –tryhard -m 1). Common sites were identified using MULTOVL (Aszodi, 2012) after identifying peaks with Model-based analysis of ChIP-Seq (MACS) (Zhang *et al*, 2008b).

### Peak calling and similarity analysis

BrdU-positive genomic regions and sororin peaks were called using the R package BayesPeak (Spyrou *et al*, 2009) version 1.18.2, with 1,000 bp bin size and corresponding control samples. Enriched bins were filtering for posterior probability > 0.5 and adjacent bins merged. Dependence of the localization of sororin peaks on the position of the BrdU peaks was assessed using IntervalStats (Chikina & Troyanskaya, 2012). The resulting *P*-value distributions of co-localization of each sororin peak with its closest BrdU peak were compared to a *P*-value distribution of randomly shuffled sororin peaks (100 rounds performed with bedtools shuffle).

### Data quantification and analysis

Signal intensities obtained in fluorescent microscopy experiments were quantified using ImageJ (imagej.nih.gov/ij). Quantifications were processed with Microsoft Excel and GraphPad Prism. Significance levels were quantified using unpaired *t*-test.

**Expanded View** for this article is available online.

### Acknowledgements

We wish to thank Julia Schmitz and Joël Beaudoin who originally analyzed the turnover of sororin on chromatin, Roman Stocsits for MACS peak calling and overlap analysis, Uta Moehle-Steinlein, Georg Petzold, Venugopal Bhaskara, Erika Schirghuber, Jason C. Bell and the laboratories of Johannes Zuber, Daniel Gerlich, Masato Kanemaki, Arabella Meixner, and Aaron Straight for providing vectors and expertise. We further thank the CSF NGS Unit (csf.ac.at) for deep sequencing, and the IMP/IMBA Biooptics facility and members of the J.-M.P. group for discussions and assistance. Research in the laboratory of J.-M.P. is supported by Boehringer Ingelheim, the Austrian Science Fund (FWF special research program SFB F34 "Chromosome Dynamics" and Wittgenstein award Z196-B20), the Austrian Research Promotion Agency (FFG, Laura Bassi Center for Optimized Structural Studies), the Vienna Science and Technology Fund (WWTF LS09-13), and the European Community's Seventh Framework Programme (FP7/2007-2013) under grant agreement 241548 (MitoSys).

### Author contributions

All authors designed experiments and interpreted data. RL performed photobleaching and RNAi experiments and analyzed cells overexpressing sororin. EK generated and analyzed *Sororin* conditional knockout mice and sororin-LAP-AID cells. MPI performed ChIP and DIP experiments. HE performed peak calling and correlation analysis. MHI-A analyzed mouse embryonic development. GAB generated and analyzed *Smc3* conditional knockout fibroblasts. GW and DAC performed experiments during the revision of the manuscript. RL and J-MP prepared the manuscript.

### Conflict of interest

The authors declare that they have no conflict of interest.

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
