## [Review Process File · The EMBO Journal]

Manuscript EMBO-2015-92532

Sororin actively maintains sister chromatid cohesion

Rene Ladurner, Emanuel Kreidl, Miroslav P Ivanov, Heinz Ekker, Maria Helena Idarraga-Amado, Georg A Busslinger, Gordana Wutz, David A Cisneros and Jan-Michael Peters

Corresponding author: Jan-Michael Peters, Research Institute of Molecular Pathology

Review timeline:

Submission date:	13 July 2015
Editorial Decision:	24 August 2015
Revision received:	18 December 2015
Acceptance:	17 January 2016
Accepted:	17 January 2016

Editor: Bernd Pulverer

Transaction Report:

1st Editorial Decision

24 August 2015

Thank you for submitting your manuscript for consideration by the EMBO Journal. It has now been seen by three referees whose comments are shown below.

Given the referees' overall positive recommendations, I would like to invite you to submit a revised version of the manuscript, addressing the comments of all three reviewers. The following specific experiments suggested by the referees are in our opinion realistic and important additions, which will render the paper more compelling.

Referee 1:

- 1) test a double alanine mutant to exclude structural defects (see also referee 2).
- 2) execute sororin destruction in G2 and M synchronized cells.
- 3) show as supplementary data FACS profiles.
- 4) add controls to fig 5D for Esc1, 2 expression, chromatin binding and cell cycle distribution; you may wish to re-run the gel to ensure Smc3 acetylation is in the linear range.

Referee 2:

- 1) provide clearer evidence if the effects of ESCO depletion on SMC3 acetylation is modest due to potential compensation between the 1 and 2 isoform.
- 2) Exclude possibility that an unexplained G2/M effect allows for sororin to stabilize cohesion (see also cell cycle experiments suggested by ref 1).

Both ref 1 and 2 question the assumption that the QQ and RR SMC3 mutants are acetylation mimetic. While ref 2 requests rewording or even deletion of the RR data, we do not feel this would address the questions raised, as the mutant generates positive data that ought not to be discounted, but rather better understood, as suggested by ref. 1. Thus, we encourage clearer experimental evidence that the mutants act in the manner proposed in the manuscript.

Referee 3:

- 1) does cohesion stabilization in G2 (fig 3G) require Wapl displacement?
- 2) advance the study to show how cohesion accumulates at firing origins.
- 3) add source data including additional micrographs and/or a quantification to fig. 5H.
- 4) show Pds5 protein turnover in G2.
- 5) add FRAP data for heterochromatin and euchromatic regions.

We also note that a key novel finding presented in your manuscript is the fact that sororin binds highly dynamically in G2 phase, indicating that chromatin bound cohesin is recognized by sororin, but the current dataset provides no specific insight into what regulates this (it cannot be merely acetylated Smc3). As referee 3 points out 'The results suggest that cohesin also tends to accumulate at firing origins. Is there any mechanism proposed for the accumulation of cohesin at the firing origins?'. Any additional insight into this would in our view render the dataset more interesting for this journal. We appreciate that the referees offer limited specific experimental suggestion in this regard beyond analyzing Pds5 turnover and heterochromatin and euchromatic regions (ref 3).

I should add that it is EMBO Journal policy to allow only a single round of revision, and acceptance of your manuscript will therefore depend on the completeness of your responses in this revised version.

When preparing your letter of response to the referees' comments, please bear in mind that this will form part of the Review Process File, and will therefore be available online to the community. For more details on our Transparent Editorial Process, please visit our website: http://emboj.emboPress.org/about#Transparent_Process

REFeree REPORTS

Referee #1:

Cohesion between sister chromatids is thought to be established by the cohesin acetyltransferases Esco1 and Esco2 in S-phase and maintained by sororin in G2/M. This manuscript provides further evidence for this hypothesis in mouse and human cells. It also provides some new insights into sororin function and chromatin binding dynamics. The authors examine the functional requirement of sororin in different cell cycle stages using a degron-tagged version of the protein, characterize its turnover on chromatin by iFRAP, and study how and when sororin is recruited to chromatin using ChIP-Seq. The result that sororin locally correlates with replicating regions is particularly interesting.

That said I have some questions, concerns, and suggestions about some of the experiments, which need to be addressed.

Major points:

1. The authors depend heavily on two lysine site Smc3 mutants (QQ and RR) that they refer to as an acetyl-mimics, based on their binding to sororin. However their data do not rule out the alternative possibility, namely that the effect of the mutations is not specific to their chemical identity (i.e., they induce a structural and functional change attributable to the loss of lysine, rather than mimicry of acetylated lysine). One way to distinguish this is to compare the effects of a more severe mutant that has no plausible chemical similarity to acetylated lysine (for example AA). If such a mutant behaves the same as the QQ and RR mutants (that is, it supports cohesion in Smc3-null MEFs and Esco1/2-depleted HeLa cells), then it would suggest that these mutations bypass the requirement for acetylation by removing lysine from this surface near the Smc3 ATPase head, rather than by mimicking the chemical attributes of acetylated lysine. If so, these mutants should be more carefully

described as acetyl-bypass mutants (not acetyl-mimics) throughout the text.

2. The authors attempt to test sororin's contribution to post-replicative "cohesion maintenance," but the experiments include many cells that were still replicating (and thus potentially still "establishing" cohesion at late-replicating genomic regions like centromeres) while the degron-tagged form of sororin was being destroyed. A clearer test would be to initiate and complete the destruction of sororin in cells that are beyond S phase entirely. G2-phase synchronization could be achieved using Cdk1 inhibitors or cyclin B depletion, while mitotic synchronization could be done with nocodazole or other checkpoint-activating drugs. From Holland et al (2012) it appears the auxin degron system is active in M phase, so the requested analysis is feasible and should resolve the question more clearly.

Minor points:

1. Some RNAi experiments suggest cross-depletion of other factors (for instance Fig. 2F, where Escol1/2 levels fall after sororin depletion) or fail to address and exclude cell cycle effects as a confounding variable (for instance Escol1/2 depletion followed by iFRAP or chromatin fractionation). FACS DNA content profiles for these experiments should be shown, in order to evaluate the efficacy of synchronization.
2. Why does Smc3 acetylation decrease after sororin inactivation (Fig. 5D; the lane is overloaded based on histone H3 levels, but the acetylated Smc3 signal is the same or lower than the control). Does this reflect a change in Escol1 or Escol2 abundance, chromatin binding, or cell cycle distribution?
3. Why are there no background bands in Figure 2F for Escol1/2 as there are in Figure 1A?
4. The sample size (number of cells analyzed) for iFRAP experiments needs to be provided.

Referee #2:

Sister chromatid cohesion is essential for eukaryotic cell division and is provided by the cohesin complex. Cohesin is regulated by pro- and anti-cohesion factors, but details of the action and interplay between these factors is insufficiently understood. Ladurner et al address this important gap, test a prevailing hypothesis, largely confirm it and add valuable and substantial new pieces of information. The focus of this study is sororin, a pro-cohesion factor, in conjunction with the acetyl transferases ESCO1, ESCO2. Two mouse and fibroblast models were generated that allow depletion of the gene or the protein, and showed revealing phenotypes.

This is an informative and sound paper, and there are only a few editorial points to consider:

In the introduction, data obtained from quite different model systems were not differentiated as such, i.e. the recommendation is to indicate where data originated from as it is not clear at all whether all processes and mechanisms are shared between species (in fact, it is clear that there are differences). For example the Uhlmann & Nasmyth paper 1998 presents data generated using *S. cerevisiae* (and sororin/dalmanian exists in invertebrates/vertebrates only).

The ESCO depletion experiments rely on RNAi with its well-known limitations. The contribution of ESCO1 to SMC3 acetylation seems to be rather low (Fig. 1A), although it is difficult to exclude that under conditions of reduced ESCO1, ESCO2 activity would be enhanced (and vice versa). Indeed, the presence of ESCO1 seems to be increased upon ESCO2 siRNA treatment. This may be worth mentioning. Nevertheless, it is clear, as the authors state, that both enzymes "contribute" to SMC3 acetylation.

How do the authors explain that a QQ and a RR mutant of K105/106 of SMC3 associate with sororin and thus QQ and RR mutants "with respect to sororin binding" both resemble acetylated SMC3? The mechanism by which sororin binds to these two very different mutants may be distinct, and binding to the RR may be through an "artificial" mechanism, which does not reflect normal sororin binding to acSMC3. I would hesitate to call the RR mutant an acetyl-mimicking mutant,

even though it behaves in FRAP (Fig. 2A, C) like wt and the QQ.

If levels of the SMC3-LAP are much lower (Fig. 2E) than endogenous wt SMC3, how can the LAP-tagged support full (?; what does "similar extent mean"?) cell viability upon elimination of the endogenous SMC3 (p. 7). Other experiments have shown that lowering SMC3 as much causes SMC1 to become unstable. The levels at which SMC3-LAP is expressed seem to be similar to levels reported in several papers upon siRNA treatment showing phenotypes. That requires explanation.

On page 7 the authors conclude: "This indicates that the only function of Esco1 and Esco2 in stabilizing cohesin on chromatin is to acetylate Smc3." Yes, but only if the RR mutant truly reflects acSMC3, which is not certain. In later experiments the RR mutant was not included anymore - is it necessary to include in the initial figures?

On page 17 the authors write "Instead, it must be some property of cohesin itself that allows sororin to bind." An alternative would be that the state of the cell cycle, not a replication or cohesin component/property but rather cyclin-CDK/associated or other factors present only in S/G2 allow sororin to stabilize cohesin - independently if its ability to bind cohesin.

To my knowledge, mouse proteins are to be written in capitals.

Referee #3:

The author's group previously demonstrated that Sororin interacts with acetylated cohesin, and Sororin activity is essential for maintaining cohesion by inhibiting Wapl function in vertebrate cells (Nishiyama et al, Cell 2010). In this manuscript, the authors have extended the functional analysis of Sororin and explore the molecular mechanisms of Sororin-dependent maintenance of sister chromatid cohesion during cell cycle. They demonstrate that the acetyltransferases Esco1 and Esco2 can stabilize cohesin on chromatin through the acetylation of Smc3 and Sororin binding. Sororin overexpression induces cohesin stabilization on chromatin at G2 phase (after DNA replication) when Smc3 is acetylated by Esco1 and Esco2. Sororin, which accumulates predominantly at early origins is highly dynamic at G2 phase, implying that the DNA replication machinery or the process itself is dispensable for Sororin interaction with cohesin. Finally, because Sororin is required for cohesion even after DNA replication, the authors conclude that Sororin is a cohesion maintenance factor.

In general, the experiments are well designed and executed in a good manner, and the results are convincing. Although the requirement of Sororin function for Wapl inhibition has already been proposed by this group, this study provides solid and advanced results about Sororin using knockout mice and the degron system targeting Sororin. I have some minor comments listed below, which should be clarified before this manuscript is accepted.

1. In Figure 3G; cohesin becomes stable in sororinKBM-Flag expressed cells at G2 phase. Is this caused by the displacement of Wapl from cohesin?
2. In Figure 4, the peaks of Sororin, BrdU and Smc3 along the chromatin are largely overlapped during early S phase. This suggests that cohesin also tends to accumulate at firing origins. Is there any mechanism proposed for the accumulation of cohesin at the firing origins? The author should discuss this point.
3. In Figure 5H, the authors show a picture of a cell with only one unaligned chromosome, and the quantification data show that ~50% of cells show misaligned chromosomes among Sororin-depleted cells. Is the picture really representative? I cannot imagine how such weak misalignment leads to the drastic increase of tetraploidy (Figure 5F). From the figure legend, I could not determine when the cells were collected and analyzed in Figure 5G, 5H and 5J.
4. The rapid turnover of Sororin is important information for understanding the regulation of cohesion. The authors' group previously reported that Sororin binds to cohesin through Pds5 (Nishiyama et al., Cell 2010). It is important to examine the turnover of Pds5 in G2 phase. Is it more similar to Smc3 (slow) or to Sororin (rapid)?

5. Regarding the turnover of Sororin (Pds5) in G2 phase (Figure 7), FRAP could be compared between heterochromatic and euchromatic regions.

1st Revision - authors' response

18 December 2015

Response to comments by Referee #1:

This referee appreciated our characterization of sororin's function in mouse and human cells and showed particular interest into our finding that sororin localization on chromatin correlates with sites of replication. However, she/he had "some questions, concerns, and suggestions about some of the experiments, which need to be addressed".

Major points:

1. *The authors depend heavily on two lysine site Smc3 mutants (QQ and RR) that they refer to as an acetyl-mimics, based on their binding to sororin. However their data do not rule out the alternative possibility, namely that the effect of the mutations is not specific to their chemical identity (i.e., they induce a structural and functional change attributable to the loss of lysine, rather than mimicry of acetylated lysine). One way to distinguish this is to compare the effects of a more severe mutant that has no plausible chemical similarity to acetylated lysine (for example AA). If such a mutant behaves the same as the QQ and RR mutants (that is, it supports cohesion in Smc3-null MEFs and Escal/2-depleted HeLa cells), then it would suggest that these mutations bypass the requirement for acetylation by removing lysine from this surface near the Smc3 ATPase head, rather than by mimicking the chemical attributes of acetylated lysine. If so, these mutants should be more carefully described as acetyl-bypass mutants (not acetyl-mimics) throughout the text.*

We thank the referee for pointing out that the term "acetyl-mimicking" can be misleading in the sense that it could be misunderstood to indicate structural similarity to acetylated lysine, in particular as the term has been used in this latter sense in the literature. However, our intention was to use this term to describe the functional similarity between SMC3-QQ and SMC3-RR mutants with acetylated SMC3, as defined by their ability to bind to sororin. However, because our terminology was potentially misleading, we have now replaced it with the term "acetyl-bypass mutants".

In addition, we have tried to analyze the behavior of SMC3-AA mutants in FRAP assays, as suggested by this and the second referee. Unfortunately, we were unable to obtain MEFs in which SMC3-AA was stably expressed from a BAC and in which *Smc3* could be deleted by expression and Cre recombinase so far, and in HeLa cells we observed that SMC3-AA expressed from a BAC turned over on chromatin in G2 phase much faster even than wild-type cohesin, with a residence time of 8.05 minutes. We currently do not know the reason for this finding but suspect that this defect is unrelated to their acetyl-bypass properties as not only SMC3-QQ but also SMC3-RR mutants behaved differently in this assays, namely like acetylated SMC3, even though arginine residues are not particularly similar in structure to acetyl-lysine residues. In other words, we would argue that SMC3-RR mutants are similar to SMC3-AA mutants insofar as neither of them is structurally very similar to acetylated SMC3. It is important to note that SMC3-RR mutants have also not been argued in the literature to structurally mimic acetylated SMC3 (this has only been proposed for SMC3-QQ and SMC3-NN), but that SMC3-RR mutants have been interpreted as non-acetylatable mutants (Ben-Shahar et al, 2008; Unal et al, 2008; Rowland et al., 2009; Sutani et al, 2009).

For these reasons, we now use the term acetyl-bypass mutants for both SMC3-QQ and SMC3-RR and have explicitly mentioned in the manuscript that SMC3-RR is not particularly similar in structure to acetylated SMC3.

2. *The authors attempt to test sororin's contribution to post-replicative "cohesion maintenance," but the experiments include many cells that were still replicating (and thus potentially still "establishing" cohesion at late-replicating genomic regions like centromeres) while the degra-tagged form of sororin was being destroyed. A clearer test would be to initiate and complete the destruction of sororin in cells that are beyond S phase entirely. G2-phase synchronization could be achieved using Cdk1 inhibitors or cyclin B depletion, while mitotic synchronization could be*

done with nocodazole or other checkpoint-activating drugs. From Holland et al (2012) it appears the auxin degron system is active in M phase, so the requested analysis is feasible and should resolve the question more clearly.

We agree with the concern of the referee regarding sister chromatid cohesion at late-replicating regions in Fig 8; however, such late-replicating regions of the genome ought to be detected by EdU labeling (since EdU was continuously present during auxin-induced degradation). The dataset in Fig 8E therefore shows sororin degradation-dependent cohesion defects exclusively in cells that had not incorporated any EdU. Furthermore, reports indicate that centromere replication is not limited to late S-phase in MEFs (Weidtkamp-Peters et al, 2006) or human cells (Erliandri et al, 2014). We nevertheless followed the referee's suggestion and induced sororin degradation in mitosis after shaking off mitotic cells and incubation with nocodazole and auxin (Fig S7). Six hours later, chromosome spreads of 56.5% of cells showed defective cohesion, whereas only 20.5% of cells treated with nocodazole but without auxin showed cohesion defects (Fig S7C), indicating that the presence of sororin is continuously required throughout mitosis.

Minor points:

3. *Some RNAi experiments suggest cross-depletion of other factors (for instance Fig. 2F, where *Esco1/2* levels fall after sororin depletion) or fail to address and exclude cell cycle effects as a confounding variable (for instance *Esco1/2* depletion followed by iFRAP or chromatin fractionation). FACS DNA content profiles for these experiments should be shown, in order to evaluate the efficacy of synchronization.*

We have now included the corresponding FACS DNA profiles (Fig S1A, S2C, S2G, S6E, S6M) for the experiments described in Fig 1, Fig 2F-G, Fig S2G-I, and Fig S6E-M. We did not observe cell cycle effects in these experiments that were detectable by FACS. However, in the interest of transparency we have now explicitly pointed out the reduction of ESCO1 levels after sororin depletion seen in Fig 2F by adding the following statement to the legend: "Note that ESCO1 immunoblot signals were also reduced in sororin depleted cells. We currently do not know if this is an effect of sororin depletion, an off-target effect or an artefact of unequal sample loading."

4. *Why does *Smc3* acetylation decrease after sororin inactivation (Fig. 5D; the lane is overloaded based on histone H3 levels, but the acetylated *Smc3* signal is the same or lower than the control). Does this reflect a change in *Esco1* or *Esco2* abundance, chromatin binding, or cell cycle distribution?*

We have repeated the experiment depicted in the original Fig 5D at two different time points (24 and 48 hours after release from serum starvation) and do not see a difference in SMC3 acetylation levels after deletion of sororin (new Fig 5D). These data are also consistent with our previous finding that sororin depletion does not affect cohesin acetylation in HeLa cells (Nishiyama et al, 2010, Fig S3C).

5. *Why are there no background bands in Figure 2F for *Esco1/2* as there are in Figure 1A?*
The unspecific bands seen for our antibodies against ESCO1 and ESCO2 in fresh antibody/blocking buffer preparations (Fig 1A) were gradually depleted from preparations over time by continuous usage. We state this observation now in the figure legend of Fig 1A: "Asterisks indicate unspecific signals that were depleted from antibody dilutions used in later experiments, presumably by repeated usage of the antibody samples."

6. *The sample size (number of cells analyzed) for iFRAP experiments needs to be provided.*

We have now included the sample size for all iFRAP and FRAP experiments in the corresponding figure legends.

Response to comments by Referee #2:

This referee felt that our study added "valuable and substantial new pieces of information" as "an informative and sound paper" with "only a few editorial points to consider."

1. *In the introduction, data obtained from quite different model systems were not differentiated as such, i.e. the recommendation is to indicate where data originated from as it is not clear at all whether all processes and mechanisms are shared between species (in fact, it is clear that there*

are differences). For example the Uhlmann & Nasmyth paper 1998 presents data generated using *S. cerevisiae* (and sororin/dalmatian exists in invertebrates/vertebrates only).

We agree with the referee that differences in the regulation of cohesion might exist between different species and have therefore now carefully reviewed our manuscript according to her/his recommendation.

The study by Uhlmann & Nasmyth 1998 was referenced in our original introduction after the sentence “Cohesin establishes cohesion during DNA replication” because we felt that it is plausible to generalize this finding to other eukaryotes, as all existing observations are indeed consistent with the possibility that cohesion is normally established during DNA replication. For example, Schmitz et al., 2007 showed by FISH that cohesion in HeLa cells clearly already exists during G2-phase, and the results reported by Tachibana-Konwalski et al., 2010 indicate that in mouse oocytes cohesion can only be established during pre-meiotic S-phase. Nevertheless, the referee is correct, and to do justice to what has been shown in the literature we have now used the following statement: “Cohesin establishes cohesion during DNA replication in yeast (Uhlmann & Nasmyth, 1998) and presumably also in mammals (Schmitz et al., 2007; Tachibana –Konwalski et al., 2010) and other eukaryotes”.

Concerning the existence of sororin in different species, we already specified in the original version of our manuscript that “In vertebrates, SMC3 acetylation results in the association of cohesin with sororin” (p. 3), and discussed later that “there may also be important differences between the mechanisms of cohesion maintenance between metazoans and other eukaryotes as orthologs of sororin have so far only been discovered in vertebrates and insects (Nishiyama et al., 2010; Rankin et al., 2005) but not in fungi and plants” (p. 20). We believe that these statements accurately describe the current knowledge in the literature. However, we feel it would be wrong to state that sororin and Dalmatian only exist in vertebrates and insects, respectively, as it cannot be excluded that orthologs (or functionally related proteins) exist in other species.

2. *The ESCO depletion experiments rely on RNAi with its well-known limitations. The contribution of ESCO1 to SMC3 acetylation seems to be rather low (Fig. 1A), although it is difficult to exclude that under conditions of reduced ESCO1, ESCO2 activity would be enhanced (and vice versa). Indeed, the presence of ESCO1 seems to be increased upon ESCO2 siRNA treatment. This may be worth mentioning. Nevertheless, it is clear, as the authors state, that both enzymes “contribute” to SMC3 acetylation.*

The referee is correctly stating that our RNAi experiments were performed under conditions that do not fully deplete ESCO1 and ESCO2 proteins. We have chosen these conditions because our initial experiments using more severe depletion conditions (longer siRNA incubation and/or prolonged cell cycle arrest) changed the cell cycle profile compared to control cells (data not shown). The conditions used in Fig 1–3 and Fig S61–M allowed us to deplete proteins within one full cell cycle (siRNA treatment at time of release from the first thymidine arrest; total incubation time with RNAi mix, 28 hours; time from treatment start to analysis, 34 hours). As a result, ESCO1, which is constitutively expressed, was more difficult to deplete than ESCO2, which oscillates during the cell cycle.

Regarding the observation of increased ESCO1 presence after ESCO2 siRNA treatment, we have now added the following sentence to the figure legend of Fig 1A: “Note that ESCO1 levels were increased on chromatin after depletion of ESCO2, raising the possibility that a decrease in chromatin bound ESCO2 might be compensated for by ESCO1 recruitment to chromatin.”

3. *How do the authors explain that a QQ and a RR mutant of K105/106 of SMC3 associate with sororin and thus QQ and RR mutants “with respect to sororin binding” both resemble acetylated SMC3? The mechanism by which sororin binds to these two very different mutants may be distinct, and binding to the RR may be through an “artificial” mechanism, which does not reflect normal sororin binding to acSMC3. I would hesitate to call the RR mutant an acetyl-mimicking mutant, even though it behaves in FRAP (Fig. 2A, C) like wt and the QQ.*

We agree with the referee that the term “SMC3 acetyl-mimicking mutant” is potentially misleading, even though we did try to explain in the original version of our manuscript that we operationally defined it by the ability of these mutants to bind sororin.

To avoid potential misunderstandings, we have now instead used the term “acetyl-bypass mutant” suggested by referee #1 (please see our response to point 1 of referee #1).

We have not addressed by which structural mechanism sororin binds to cohesin containing SMC3-QQ or SMC3-RR mutations, which is not even well understood for wild-type sororin. However, for the following reasons it is difficult to imagine that SMC3-RR binds sororin through an

“artificial” mechanism: We show in this study that the sororin gene (*Cdca5*) is essential for viability in the mouse, consistent with the previous observation that sororin is essential for sister chromatid cohesion (Rankin et al., 2005; Schmitz et al., 2007), which is also known to be essential for viability. The simplest interpretation of these observations and our previous finding that sororin is only required for cohesion in the presence of WAPL (Nishiyama et al., 2010) is that sororin is essential for viability because it is required for cohesion. Nevertheless, MEFs only expressing SMC3-RR but not endogenous SMC3 are viable, as we report in this study. This indicates very strongly that the SMC3-RR mutant must be able to interact with sororin in a functional manner, i.e. presumably in a manner which is very similar to how acetylated SMC3 interacts with sororin. This notion is also supported by our previous finding that cohesin complexes containing wild-type SMC3, SMC3-QQ and SMC3-RR all interact with the same cohesin subunits (as measured by mass spectrometry), including PDS5 proteins which bind sororin and are required for sororin-cohesin interactions (Nishiyama et al., 2010; Fig S3G).

4. *If levels of the SMC3-LAP are much lower (Fig. 2E) than endogenous wt SMC3, how can the LAP-tagged support full (?; what does "similar extent mean"?) cell viability upon elimination of the endogenous SMC3 (p. 7). Other experiments have shown that lowering SMC3 as much causes SMC1 to become unstable. The levels at which SMC3-LAP is expressed seem to be similar to levels reported in several papers upon siRNA treatment showing phenotypes. That requires explanation.*

Our observation that SMC3-LAP is expressed at lower levels than endogenous SMC3 (Fig. 2E, Fig. 2A, Fig. 3C), yet that MEFs only expressing SMC3-LAP are viable, implies that these cells can live with cohesin levels that are lower than the ones normally seen in MEFs. This finding is in excellent agreement with previous studies which reported that budding yeast can live with cohesin levels as low as 13% compared to wild-type levels (Heidinger-Pauli et al., 2010), and that reduction of SMC3 levels in HeLa cells by RNA interference to 35-50% did also not prevent cell division (Laugsch et al., 2013).

We have now rephrased our original statement “forms of SMC3-LAP supported cell viability to a similar extent” to explain more clearly what we observed: “Upon Cre-mediated deletion of endogenous *Smc3*, cells without SMC3-LAP stopped to proliferate, whereas cells containing wild-type SMC3-LAP, SMC3-LAP(QQ) or SMC3-LAP(RR) continued proliferation at a comparable rate.”

Since it is well known (since the discovery of degradation of excess globin polypeptide chains in thalassemias) that the abundance of one protein can have effects on the stability of its binding partners, and such an effect has been reported to exist for SMC1 and SMC3 by Laugsch et al., 2010, we would expect that SMC1 levels are also reduced in the MEFs expressing only SMC3-LAP. However, since our complementation experiment was merely intended to reveal potential differences between wild-type and acetyl-bypass mutant SMC3-LAP, which we did not detect, we did not analyze if SMC1 levels were also reduced in cells expressing only SMC3-LAP.

5. *On page 7 the authors conclude: "This indicates that the only function of Esco1 and Esco2 in stabilizing cohesin on chromatin is to acetylate Smc3." Yes, but only if the RR mutant truly reflects acSMC3, which is not certain. In later experiments the RR mutant was not included anymore - is it necessary to include in the initial figures?*

In our view also the experiments with the SMC3-RR mutant fully support the above mentioned conclusion because this mutant bypasses the requirement for ESCO1 and ESCO2 in terms of cohesin stabilization on chromatin in G2-phase to the same extent as the SMC3-QQ mutant (Fig. S2G-I). Furthermore, as explained under point 2 above, there are strong reasons to believe that the SMC3-RR mutant must interact with sororin in a functional manner, i.e. presumably in a manner similar to how acetylate SMC3 interacts with sororin

We believe that it is important to include the results obtained with the SMC3-RR mutant because this mutant does not structurally resemble acetylated SMC3 and is therefore an acetyl-bypass mutant in the sense discussed in our reply to point 1 of referee #1.

We did not include the RR mutant in later tests (Fig 3) because these experiments only address the requirement for sororin recruitment to chromatin (and not the properties of different SMC3 mutants), and the QQ mutant seemed sufficient as a model for a cohesin acetylation bypass in G1-phase.

6. *On page 17 the authors write "Instead, it must be some property of cohesin itself that allows sororin to bind." An alternative would be that the state of the cell cycle, not a replication or*

cohesin component/property but rather cyclin-CDK/associated or other factors present only in S/G2 allow sororin to stabilize cohesin - independently if its ability to bind cohesin.

We had concluded that “it must be some property of cohesin itself that allows sororin to bind” because we had found that soluble cohesin-SMC3-QQ could bind sororin very well not only in G2 but also in G1 phase, arguing against the possibility that the cell cycle state *per se* could control sororin-cohesin interactions.

We had not considered the more complicated possibility, suggested by this referee, that not the binding of sororin to cohesin but sororin’s ability to stabilize cohesin on chromatin could be cell cycle regulated. We did not consider this to be a very plausible possibility because it is already clear that the binding of sororin to cohesin is regulated, and that this step is principally sufficient to explain why cohesin becomes stabilized on chromatin only from DNA replication onwards. In other words, there was no good reason to postulate yet another regulatory step.

However, the referee is correct that the existence of such a step cannot be excluded. We therefore performed an experiment to address the possibility that CDK activity enables cohesin stabilization by sororin. The results obtained are shown in the new Fig. S3D–F. In this experiment, we synchronized cells in G2-phase by double thymidine arrest-release, transfected these cells with sororin in the presence of the CDK inhibitor RO-3306 and analyzed cells by iFRAP 14 hours later while they were arrested in G2-phase due to CDK1 inhibition. We found that overexpressed sororin could still stabilize cohesin while CDK activity was inhibited, arguing against the possibility that CDK or CDK-dependent activities regulate sororin-dependent cohesin stabilization.

7. To my knowledge, mouse proteins are to be written in capitals.

We thank the referee for pointing this out; all gene and protein names now follow HGNC and MGI guidelines.

Response to comments by Referee #3:

This referee expressed that our “experiments are well designed and executed in a good manner, and the results are convincing” and that “this study provides solid and advanced results”. She/he further had “some minor comments listed below, which should be clarified before this manuscript is accepted”.

1. In Figure 3G; cohesin becomes stable in sororinKBM-Flag expressed cells at G2 phase. Is this caused by the displacement of Wapl from cohesin?

Our previous work showed that *in vitro* sororin can displace WAPL from its binding partner PDS5A (Nishiyama et al, 2010). However, evidence published in the same paper suggests that in the presence of excess amounts of sororin, WAPL is not displaced from cohesin (Figure S4F), presumably because it also binds to cohesin via SA1 or SA2 (Shintomi et al, 2009; Rowland et al, 2009).

Using FRAP, we have now tested the turnover of GFP-tagged WAPL on chromatin in G2-phase (Fig S6E–H) and determined a mean residence time on chromatin of 46 seconds. We also tested if depletion of sororin would allow WAPL to remain bound to chromatin for a longer time (Fig S6I–M). This experiment revealed a very small increase in chromatin residence time (54 seconds after control depletion, 61 seconds after sororin depletion), but this increase was statistically not significant ($p = 0.615$), arguing that sororin does not cause changes in WAPL-cohesin interactions that are detectable in our assays.

2. In Figure 4, the peaks of Sororin, BrdU and Smc3 along the chromatin are largely overlapped during early S phase. This suggests that cohesin also tends to accumulate at firing origins. Is there any mechanism proposed for the accumulation of cohesin at the firing origins? The author should discuss this point.

We thank the referee for making this observation. We have now analyzed the correlation of SMC3 peaks with BrdU-incorporating regions in early S-phase and could confirm a non-random accumulation of a subset of cohesin peaks at early replicating sites (Fig 4D), although not nearly to the same degree as sororin correlated with these sites (Fig 4C). As expected (because cohesin binds to the same sites throughout the cell cycle – see below), also SMC3 peaks in G2-phase correlated with the same early replicating sites (Fig 4D). The observation that most cohesin binding sites do not change between G1 and G2-phase (Wendt et al, 2008), and between early S-phase and G2-phase (this study), indicates that cohesin positioning does not generally depend on DNA replication.

However, the inverse could be true, that DNA replication is preferentially initiated at certain cohesin sites, as suggested by Guillou et al., 2010. Another possible explanation is that subsets of cohesin and of early replicating sites correlate due to their preference for high chromatin accessibility; given that actively transcribed regions of the genome are replicating early (Gilbert 2002), and a fraction of cohesin binding sites is preferentially found at transcription start sites (Misulovin et al, 2008; Parelho et al, 2008). We did not explore these possibilities further because we felt that such experiments are beyond the scope of our study, but now show and mention the observed co-occurrence of SMC3 binding sites and sites of early DNA replication.

3. *In Figure 5H, the authors show a picture of a cell with only one unaligned chromosome, and the quantification data show that ~50% of cells show misaligned chromosomes among Sororin-depleted cells. Is the picture really representative? I cannot imagine how such weak misalignment leads to the drastic increase of tetraploidy (Figure 5F). From the figure legend, I could not determine when the cells were collected and analyzed in Figure 5G, 5H and 5J.*

We apologize for not specifying these important parameters earlier. We now indicate in the figures and in the figure legend of our revised manuscript at which time points these assays were performed. The weak chromosome misalignment phenotype after *Cdca5* deletion was detected within 36–60 hours after release from starvation (original Fig 5H, now Fig 5G), while severe changes in nuclear morphology were seen only 5–7 days after release from starvation (original Fig 5G, now Fig 5H). The immunoblots and chromosome spread analyses in Fig 5I–J were performed 24 hours after deleting the *Cdca5* gene using immortalized mouse fibroblasts as stated in the figure legend. Furthermore, we have included 6 more micrographs as Figure S5C to show the chromosome misalignment after deleting *Cdca5* compared to 6 micrographs from cells treated without tamoxifen.

4. *The rapid turnover of Sororin is important information for understanding the regulation of cohesion. The authors' group previously reported that Sororin binds to cohesin through Pds5 (Nishiyama et al., Cell 2010). It is important to examine the turnover of Pds5 in G2 phase. Is it more similar to Smc3 (slow) or to Sororin (rapid)?*

We agree that this is an interesting experiment and have performed FRAP on HeLa cells expressing GFP-tagged PDS5A from its endogenous locus, and compared these data to a similarly generated GFP-sororin and a GFP-WAPL HeLa cell line (Fig S6E–H). This analysis revealed that also WAPL and PDS5A interact with G2-phase chromatin in a highly dynamic manner, with WAPL showing the fastest and sororin the slowest turnover on chromatin. These new results imply that the stability of cohesin on DNA is regulated by proteins which themselves interact with cohesin in a highly dynamic manner.

5. *Regarding the turnover of Sororin (Pds5) in G2 phase (Figure 7), FRAP could be compared between heterochromatic and euchromatic regions.*

We have analyzed FRAP at regions containing sororin foci at chromocenters and compared these to regions without such foci (Fig S6A–D). By curve fitting FRAP data we found that the relative amount of chromatin-bound sororin is significantly increased at heterochromatic foci, indicating that the chromatin environment also influences sororin's binding to chromatin.

Accepted

17 January 2016

All three referees have now commented on the manuscript again and endorse publication. I am therefore very pleased to inform you that your manuscript has been accepted for publication in the EMBO Journal.

I would like to recommend a few cosmetic changes to ensure key information is maximally accessible to the readership:

- 1) please integrate the materials and methods into the main paper (we do not have truncated in paper methods like Nature and others).
- 2) Please also integrate most or all of the supplementary citations - they are not integrated into bibliographic databases as supplementary material and less visible. If any references are non-essential, please remove. Please also note the key references Minamino et al, Curr Biol, June 2015 pointed out by one of the referees, which should be discussed and cited.
- 3) we now have expandable figures (EV) embedded within the main paper so that key data are more visible. Can you please render at least 4 for your supplementary figures 'expandable view' figures

and cite them in the text as 'EV1, 2, etc'. Instructions on the nomenclature are in the journals 'guide to authors'.

Please ensure that you return all the relevant forms outlined below to avoid delays in production.

Thanks you for you synopsis and cover art - this all looks excellent , as does the title and abstract. Thank you for the detailed author checklist.

Please note that we now mandate that all corresponding authors list an ORCID digital identifier. This takes <90 seconds to complete and we encourage all authors to supply an ORCID identifier, which will be linked to their name for unambiguous name identification.

REFEREE REPORTS

Referee #1:

I was satisfied by the authors' changes in the revised manuscript. Hopefully this allows to proceed with your editorial decision.

Referee #2:

The authors have appropriately addressed all my previous points, which were mostly editorial, but the authors have also added important new data. I think that some of the messages of this paper now are conveyed in a more precise fashion and thus the paper is improved. I have no further concerns. One minor recommendation is to cite the recent paper by Minamino et al, Curr Biol, June 2015, which points out differences between ESCO1 and ESCO2. That may be relevant for considerations concerning potential cross-compensation by these enzymes.

Referee #3:

The authors have properly addressed all my concerns. I think think that the manuscript is now acceptable.

Corresponding Author Name: Jan-Michael Peters

Journal Submitted to: The EMBO Journal

Manuscript Number: EMBOJ-2015-92532